# Hyper: Hyperparameter Robust Efficient Exploration in Reinforcement Learning

Yiran Wang [* 1]   Chenshu Liu [* 2]   Yunfan Li [1]   Sanae Amani [1]   Bolei Zhou [3]   Lin F. Yang [1]

## Abstract

The exploration & exploitation dilemma poses significant challenges in reinforcement learning (RL). Recently, curiosity-based exploration methods achieved great success in tackling hard-exploration problems. However, they necessitate extensive hyperparameter tuning on different environments, which heavily limits the applicability and accessibility of this line of methods. In this paper, we characterize this problem via analysis of the agent behavior, concluding the fundamental difficulty of choosing a proper hyperparameter. We then identify the difficulty and the instability of the optimization when the agent learns with curiosity. We propose our method, hyperparameter robust exploration (**Hyper**), which extensively mitigates the problem by effectively regularizing the visitation of the exploration and decoupling the exploitation to ensure stable training. We theoretically justify that **Hyper** is provably efficient under function approximation setting and empirically demonstrate its appealing performance and robustness in various environments.

## 1. Introduction

Reinforcement learning (RL) is a paradigm that solves sequential decision-making problems by maximizing the expected cumulative reward $r$, which is composed of both explorations, the process of the agent discovering new features from the environment, and exploitations, the process by which the agent learns to tackle the task using the knowledge already gained. Despite the astonishing success achieved with this paradigm (Mnih et al., 2013; 2015; Silver et al.,

*Equal contribution  [1]Department of Electrical & Computer Engineering, University of California, Los Angeles, California, USA [2]Terasaki Institute for Biomedical Innovation, Los Angeles, California, USA [3]Department of Computer Science, University of California, Los Angeles, California, USA. Correspondence to: Yiran Wang <yiranwang1027@ucla.edu>.

*Proceedings of the 42nd International Conference on Machine Learning*, Vancouver, Canada. PMLR 267, 2025. Copyright 2025 by the author(s).

2016; 2017; Berner et al., 2019; Arulkumaran et al., 2019), RL is vulnerable to sub-optimal policies when presented with insufficient reward signals. To resolve this dilemma, curiosity-driven exploration methods provide a promising solution (Bellemare et al., 2016; Pathak et al., 2017; Ostrovski et al., 2017; Burda et al., 2018; Machado et al., 2020; Pathak et al., 2019).

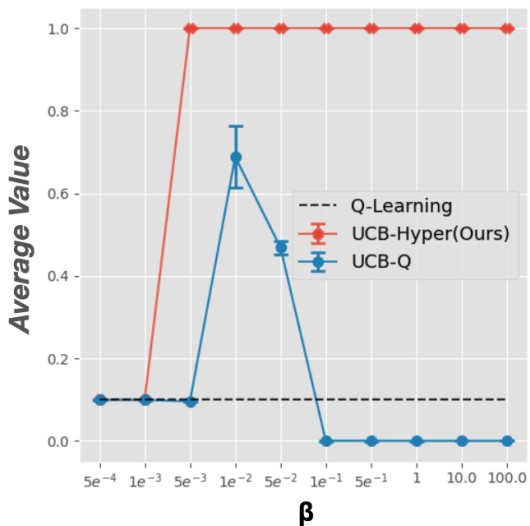

*Figure 1.* Performance of pure exploitation, curiosity-driven exploration, and our algorithm with different choices of $\beta$, each data point is the averaged performance after 1M steps training over 5 runs. Curiosity-driven (UCB-Q) is very sensitive to hyperparameter $\beta$. We propose **Hyper**, which is empirically robust to $\beta$, and theoretically efficient.

Curiosity-based algorithms, which originate from solving the bandit problem (Auer, 2002), enhance the exploration by intrinsically rewarding the agent for exploring the environment. These algorithms optimize the policy $\pi$ to maximize the joint reward function $r + \beta b$, where $r$ is the explicit task reward, $b$ is the intrinsic reward for exploration, and $\beta$ is the curiosity hyperparameter that scales the intrinsic reward. When designing curiosity-based algorithms, the intrinsic reward is often quantized by the uncertainty of the transition that the agent takes (Bellemare et al., 2016; Azar et al., 2017; Jin et al., 2018; Yang & Wang, 2020; Jin et al., 2020). The

agent is then encouraged to take uncertain transitions and explore the whole environment besides maximizing the task reward (Bellemare et al., 2016). As the agent visits a transition more often, the intrinsic reward decays and eventually becomes negligible, thereby making the extrinsic reward dominate the joint reward function, and the agent can then learn to exploit.

Curiosity-based algorithms have achieved great success in exploration-intensive problems in recent advancements, which supersedes all previous RL algorithms that solely maximize task reward (Bellemare et al., 2016; Burda et al., 2018; Pathak et al., 2019), and is proven to be theoretically efficient, according to (Auer, 2002). However, their empirical efficiency is highly dependent on the choice of hyperparameter $\beta$. The intricate relationship between the intrinsic and extrinsic reward has an important impact on the behavior of the agent. A small change of $\beta$, which controls the relative weight of the intrinsic reward $b$ and the task reward $r$, would heavily change such a relationship, leading to the sensitivity of this coefficient.

Specifically, when the cumulative intrinsic reward dominates the extrinsic one, it causes over-exploration, i.e. the agent will keep visiting uncertain transitions instead of exploiting the task. Conversely, if the intrinsic reward is too small to encourage the agent to change from its current policy, the algorithm will likely yield a sub-optimal result. Furthermore, large intrinsic rewards will cause problems in the optimization process by introducing a large bias to the fitting of the neural networks thus making the learning process hard to converge (Schäfer et al., 2021; Whitney et al., 2021).

Predominantly, existing curiosity-based algorithms exhibit a conservative inclination towards small $\beta$. A higher value of $\beta$ would heavily corrupt the task if not carefully tuned for different environments. The involvement of the neural network in practice makes the curiosity-driven methods suffer from optimization instability due to frequent policy changes. This limits the existing methods from using large $\beta$ to sufficiently explore the environment and increases the chance of getting stuck in the sub-optimal policy.

To this end, we propose our algorithm **Hyper** to solve the problem by controlling the visitation distribution of the agent exploration and mitigate the optimization instability by increasing the persistence of the visitation for adopting large $\beta$. **Hyper** is theoretically efficient, and it leverages an additional policy to decouple the exploitation learning from the exploration to prevent over-exploration in practice. It regularizes the visitation distribution of the agent's exploration to increase the exploration persistence, mitigating the optimization instability caused by frequent policy changes.

Our contributions are summarized as follows:

- We identify the sensitivity of the curiosity-driven exploration to the coefficient $\beta$ through a delicately designed example, and accordingly design a novel algorithm **Hyper** to resolve the challenges.

- We theoretically justify the efficiency of **Hyper** through rigorous analysis, and empirically demonstrate **Hyper** performs comparable, even favorable performance compared to both exploration and exploitation policies.

- We empirically analyze the robustness of **Hyper** to $\beta$, where **Hyper** shows substantially lower sensitivity compared to the original curiosity-driven exploration algorithm.

## 2. Preliminaries

In this paper, we formulate the RL problem as an episodic Markov Decision Process (MDP) (Bellman, 1957) under episodic setting, denoted by $(\mathcal{S}, \mathcal{A}, H, \gamma, r, \mathbb{P})$, where $\mathcal{S}$ is the state space, $\mathcal{A}$ is the action space, $\mathbb{P} = \{P_h\}_{h=1}^H$ are the transition measures that govern the dynamics of the environment, $r_h(s, a)$ is reward function at step $h$, $H$ is the episode length. $\gamma \in (0, 1]$ is the discount factor. The task of the agent is to learn a policy $\pi : \mathcal{S} \to \mathcal{A}$ to maximize the discounted total reward $Q^\pi = \mathbb{E}_\pi[\sum_{h=1}^H \gamma^{h-1} r_h(s_h, a_h)]$. We also denote the intrinsic reward as $b(s, a, s')$, where $(s, a, s')$ is a transition. The curiosity-driven exploration method generally aims to learn a policy that maximizes $\mathbb{E}_\pi[\sum_{h=1}^H \gamma^{h-1} r_h(s_h, a_h) + b_h(s_h, a_h, s_{h+1})]$ instead. We slightly abuse the notation by omitting the sub-scripts, denoting $r_h, b_h, s_h, a_h, s_{h+1}$ as $r, b, s, a, s'$ when the context is clear.

## 3. Warm-up Example

Curiosity-driven exploration algorithms introduce a bonus term that reflects the novelty of the current transition, into the RL objective $Q^\pi = \mathbb{E}_\pi[\sum_{i=1}^H \gamma^{i-1}(r(s, a) + \beta b(s, a, s'))]$ to reward the agent for visiting novel transitions. We can decompose the joint objective into $Q^\pi = \mathcal{E}(\pi) + \beta \mathcal{I}(\pi)$, where:

$$\mathcal{E}(\pi) = \mathbb{E}_\pi[\sum_{h=1}^H \gamma^{h-1} r(s, a)]$$

$$\mathcal{I}(\pi) = \mathbb{E}_\pi[\sum_{h=1}^H \gamma^{h-1} b(s, a, s')]$$

Intuitively, the intrinsic reward coefficient $\beta$ affects how the agent balances the exploration and exploitation by scaling the value of the intrinsic value $\mathcal{I}$. When $\mathcal{E}(\pi) > \beta \mathcal{I}(\pi)$, the agent will tend to optimize the task return, hence exploiting more. Conversely, if $\beta \mathcal{I}(\pi) > \mathcal{E}(\pi)$, the agent will

tend to explore more. The curiosity-driven algorithm works ideally when the intrinsic objective $\beta\mathcal{I}(\pi)$ is initially large enough to help the agent escape the sub-optimal policy, and it gradually decreases and diminishes, then the agent can quickly learn to exploit the diverse dataset and enjoy an overall better sample efficiency (Yang & Wang, 2020; Jin et al., 2020). However, in practice, only a limited range of $\beta$ can achieve this ideal case. The sensitivity to $\beta$ of the curiosity-driven exploration comes from various aspects, in this section, we analyze this phenomenon and decompose the mechanism behind it.

We first examine the behavior of curiosity-driven exploration methods with different choices of $\beta$ from a wide range. We consider a navigation task in a **30x30** room, which only consists finite number of states and four actions taking the agent in four different directions: up, right, down left, the layout is shown in Figure 2(a). The agent starts from a fixed initial location at the center colored by blue, and the agent should find and consistently reach the optimal goal location at the lower-right corner colored by green and receive a large reward of $R$. Besides the optimal goal, there is also a sub-optimal goal that lies near the center of the room colored by purple, which will provide a small reward of $r$. Each episode will end when the horizon is met, or the agent reaches either goal. This tabular environment allows us to train the agent without the involvement of function approximation error, which allows us to isolate the behavior analysis of the curiosity-driven exploration methods. The visitation frequencies from UCB-Q agent with different value of $\beta$ are shown in Figure 2, where the grid with brighter color means more frequent visitation.

| Parameter | $\beta$ range |
|---|---|
| $R = 1.0, r = 0.1$ | $\{5e^{-3}, 1e^{-2}\}$ |
| $R = 0.1, r = 0.02$ | $\{5e^{-3}\}$ |
| $R = 10.0, r = 1.0$ | $\{1e^{-1}, 5e^{-1}\}$ |
| $R = 100.0, r = 2.0$ | $\{5e^{-1}, 1\}$ |

*Table 1.* Range of proper $\beta$ for different environment parameter

We use UCB-Q (Jin et al., 2018) to train agents in this environment, which is built upon Q-Learning (Watkins & Dayan, 1992) and leverages upper confidence bound (UCB) (Auer, 2002; Azar et al., 2017; Jin et al., 2018) as the intrinsic reward to encourage exploration. Ideally, the UCB-Q agent should sufficiently explore the room without being trapped by the sub-optimal goal, and as the intrinsic reward shrinks, it can consistently reach the optimal goal. We run UCB-Q agents with $\beta \in \{5e^{-4}, 1e^{-3}, 5e^{-3}, 1e^{-2}, 5e^{-2}, 1e^{-1}, 5e^{-1}, 1, 10, 100\}$ in the environment with a set-up of $R = 1.0, r = 0.1$ for 1 million steps and record their final performance. As shown in Figure 1, only the choice of $\beta \in [0.005, 0.5)$ can both efficiently escape sub-optimality and consistently reach the

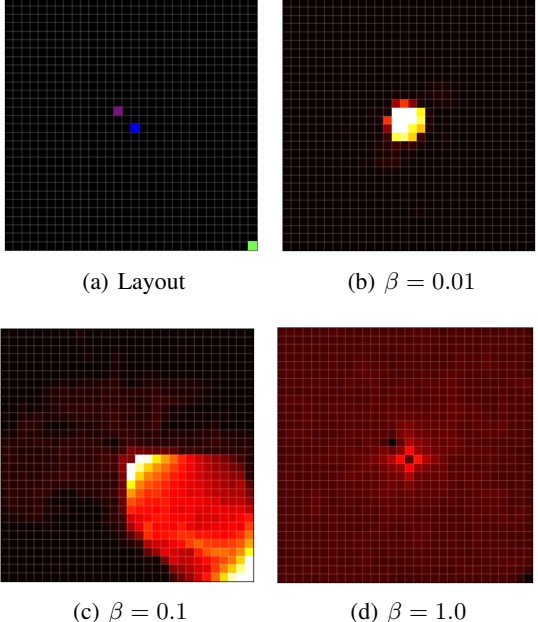

(a) Layout  (b) $\beta = 0.01$

(c) $\beta = 0.1$  (d) $\beta = 1.0$

*Figure 2.* Comparison of visitation of UCB-Q agent with different exploration coefficient, in the environment with suboptimal goal, optimal goal. Higher visitation is shown in brighter colors (visitation frequency: black < red < yellow < white). (a) Layout of the environment (b) State visitation of UCB-Q with $\beta = 0.01$, Agent gets stuck in sub-optimal policy due to insufficient exploration bonus. (c): State visitation of UCB-Q with $\beta = 0.1$, the agent finds a near-optimal policy. (d): State visitation of UCB-Q with $\beta = 1.0$, the agent over-explores and cannot learn to exploit due to the value of curiosity bonus is too high.

optimal goal.

When $\beta$ is too small, once the agent finds the suboptimal goal, the intrinsic reward is too low to encourage the agent to explore other states and escape from this sub-optimal policy, as shown in Figure 2(b). But if we choose an improperly large $\beta$, $\beta\mathcal{I}(\pi)$ dominates $\mathcal{E}(\pi)$ throughout the training, as demonstrated in Figure 2(d), the UCB-Q agent will almost uniformly visit each state. Despite the agent finding the optimal goal location, it fails to consistently revisit it even after 1 million training steps.

Unfortunately, such a proper range of $\beta$ highly depends on the configuration of the environment, in our case, the values of $R$ and $r$ determine the proper range of $\beta$. As shown in Table 1, the proper ranges are drastically different with different configurations. Hence, it is almost always necessary to perform a comprehensive hyperparameter sweep on $\beta$ to balance exploration & exploitation, which limits the applicability of this exploration paradigm.

# 4. Hyper: Algorithm and Theoretical Results

In this section, we present an overview of the Hyper algorithm, which includes both the provably efficient version (Algorithm 2) and the empirically efficient implementation (Algorithm 1). The difference between the two different versions of Hyper is that they adopt different function approximation methods. The provably efficient algorithm adopts a linear function approximation for us to examine the theoretical sample efficiency and derive a formal worst-case upper bound. The empirically efficient version is designed to work with more expressive function approximations (e.g., neural networks) and has practical enhancements that improve sample efficiency in real-world applications. The two versions share a core mechanism of "repositioning and exploration," which is introduced in the following and will be further elaborated in the next section.

The fundamental design principle behind Hyper is to decouple exploration and exploitation using a two-phase process: repositioning the agent based on exploitation knowledge and exploration based on curiosity-driven methods. By repositioning the agent to states where it can gather more useful exploratory data, Hyper ensures that exploration is directed and more persistent. This improves both the stability of the learning process and the agent's ability to avoid suboptimal policies.

The core design choice is this repositioning-and-exploration mechanism, which helps stabilize the learning process while still benefiting from curiosity-driven exploration. Instead of purely relying on intrinsic rewards to guide exploration, Hyper uses the exploitation policy to reposition the agent to promising areas of the state space. After repositioning, the agent then explores its environment to discover novel transitions. This helps mitigate the instability that typically arises from purely curiosity-driven approaches, where intrinsic rewards may dominate and lead to over-exploration or slow convergence.

**Assumption 4.1.** (Linear MDP, e.g., (Yang & Wang, 2019; 2020; Jin et al., 2020)). $\text{MDP}(\mathcal{S}, \mathcal{A}, H, \mathbb{P}, r)$ is a linear MDP whose transition $\mathbb{P} := \{\mathbb{P}_h\}_{h=1}^{H}$ is not necessarily stationary. With a feature map $\phi : \mathcal{S} \times \mathcal{A} \to \mathbb{R}^d$, such that for for any $h \in [H]$, there exists d unknown measures $\mu_h = (\mu_h^{(1)}, \mu_h^{(2)}, \mu_h^{(3)}, ..., \mu_h^{(d)})$ over $\mathcal{S}$ and an unknown vector $\theta_h \in \mathbb{R}^d$, such that for any $(s, a) \in \mathcal{S} \times \mathcal{A}$ we have:

$$\mathbb{P}_h(\cdot|s,a) = \phi(s,a)^T \mu_h(\cdot), \quad r_h(s,a) = \phi(s,a)^T \theta_h \tag{1}$$

Without loss of generality, we also assume that $\|\phi(s,a)\| \leq 1$, and $\max\{\|\mu_h(\mathcal{S})\|, \|\theta_h\|\} \leq \sqrt{d}$ for all $(s, a, h) \in \mathcal{S} \times \mathcal{A} \times [H]$

## 4.1. Theoretical Result

We now present the theoretical results that demonstrate the efficiency of the Hyper algorithm under the linear function approximation framework, whose assumption is described in Assumption 4.1. Under this framework, we form the provably efficient Linear-UCB-Hyper algorithm in Algorithm 2. The main result is that Linear-UCB-Hyper achieves provable efficiency in exploration and guarantees convergence to a near-optimal policy with high probability.

We adopt linear function approximation (Yang & Wang, 2019; 2020; Jin et al., 2020) to derive the bound in our analysis. The main theorem (Theorem 4.2) states that under this approximation, Hyper can achieve sample-efficient exploration with polynomial complexity. This guarantees that the algorithm will converge to an optimal or near-optimal policy, even in challenging exploration environments. Specifically, the result provides convergence guarantees in the worst-case scenario, where task reward signals may be sparse or difficult to access. We defer the formal proof to the appendix.

**Theorem 4.2.** *With any truncation probability $p \in (0,1)$ for the repositioning phase, it takes at most $\tilde{\mathcal{O}}(\frac{d^3 H^4}{\epsilon^2})$ steps for **Hyper** to obtain an $\epsilon$-optimal exploitation policy $\mu$ with high probability under assumption 4.1.*

While traditional curiosity-driven exploration algorithms can also achieve this worst-case upper bound, the key improvement introduced by Hyper lies in the repositioning-and-exploration mechanism. This mechanism significantly boosts the sample efficiency of Hyper compared to standard approaches by guiding exploration more effectively, thus reducing the likelihood of suboptimal exploration or unnecessary revisiting of uninformative states. This distinction is crucial and is further demonstrated in the empirical results in later sections.

# 5. Repositioning & Exploration

In this section, we dive deeper into the design choices behind the repositioning-and-exploration mechanism, which is central to Hyper's efficiency as well as robustness against $\beta$. The goal of this mechanism is to decouple task learning from exploration while regularizing exploration to ensure it remains both persistent and efficient. This decoupling allows the exploitation policy to focus on refining task-specific performance while the exploration policy is responsible for discovering novel states in the environment. We want to point out that Hyper shown in Algorithm 1 is a generic algorithm that can work with any off-policy reinforcement learning algorithms and any curiosity methods. As we will see Hyper performs well with TD3 (Fujimoto et al., 2018) as the learning algorithm and Disagreement (Pathak et al., 2019) as the curiosity module in Section 6, and it also performs well with DQN (Mnih et al., 2013) along with RND

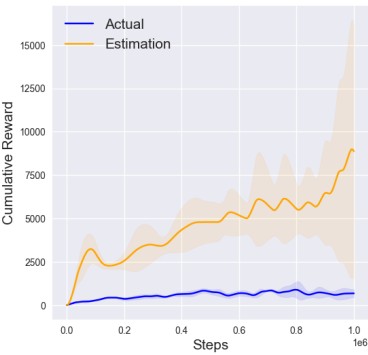

*Figure 3.* Decoupling causes distribution-shift, where the exploitation policy drastically overestimates its value (yellow), yielding poor performance (blue).

(Burda et al., 2018) in the Appendix A.5

## 5.1. Isolation of Task Learning & Regularization Over Exploration Visitation

The decision to decouple the learning of task rewards from exploration was driven by the instability observed when using large curiosity coefficients ($\beta$) in traditional algorithms. In such cases, intrinsic rewards often overpower task rewards, leading to over-exploration and poor task performance. To address this, Hyper employs a repositioning phase before each exploration phase. In the repositioning phase, the agent acts according to the exploitation policy, moving it to promising states where task learning is most likely to be effective.

This approach prevents the exploration policy from wandering too far from regions where the agent has already gained useful task knowledge, avoiding the risk of over-exploration. Additionally, by regularizing the exploration visitation—ensuring that exploration is conducted from states where the agent has already gathered substantial task-related experience—Hyper reduces the instability caused by frequent policy changes in traditional curiosity-driven methods.

This combination of isolation and regularization mitigates the distribution shift problem often observed in decoupled methods that separate task learning from exploration. Without repositioning, the exploitation policy can become detached from the data collected during exploration, leading to overestimation of value functions and poor performance in real-world tasks. Hyper addresses this by aligning the data collection process between exploration and exploitation, ensuring more reliable value estimates.

## 5.2. Truncation Probability $p$

The length of the repositioning phase is crucial in determining how far the agent is moved before exploration begins.

---

**Algorithm 1** Empirically Efficient **Hyper**

1: **Initialize:** replay buffer $D$, exploration policy $\pi$, exploitation policy $\mu$, truncation probability $p$
2: **Optional:** Linear decay schedule of $p$
3: **for** Training iteration $k = 1, 2, ..., K$ **do**
4:    repositioning_length $\leftarrow$ bounded_geom$(p, H)$
5:    **for** $i = 1, 2, ..., H$ **do**
6:      **if** $i <$ repositioning_length **then**
7:        # Repositioning phase
8:        Use $\pi$ to step in the environment
9:      **else**
10:       # Exploration phase
11:       Use $\mu$ to step in the environment
12:      **end if**
13:    Store the transition in buffer $D$
14:    # Policy Improvement
15:    Sample a batch of data $\mathcal{B} \sim D$
16:    Update $\pi$ with intrinsic reward $b$ using $\mathcal{B}$
17:    Update $\mu$ without intrinsic reward $b$ using $\mathcal{B}$
18:    Update $b$ using $\mathcal{B}$
19:    **end for**
20:    Decay $p$ based on the decay schedule
21: **end for**

---

We adopt a truncation probability $p$ that controls the stopping point for the repositioning phase, striking a balance between exploration and exploitation. Early in the training process, repositioning should not move the agent too far from regions where it has gained task knowledge. To achieve this, we initially set $p = 1 - \gamma$, where $\gamma$ is the discount factor. This value aligns the repositioning phase with the effective planning horizon of the agent's exploitation policy, ensuring that repositioning is meaningful without over-reliance on untested areas of the environment.

As training progresses, the exploitation policy becomes more stable and better at identifying high-reward regions of the environment. Hence, we gradually increase the length of the repositioning phase by decaying the truncation probability. This allows the agent to explore more distant areas of the state space as it becomes more confident in its task-specific knowledge, further improving the exploration process.

## 5.3. Truncating the Geometric Distribution

To ensure a balanced exploration process throughout training, we also truncate the geometric distribution used to sample the repositioning phase length. Specifically, in an episodic setting, sampling directly from a geometric distribution can lead to disproportionately long repositioning phases, particularly in environments with shorter horizons. Consider a setting that horizon $H = 200$, discount factor $\gamma = 0.99$, where the truncation probability will decay from $p = 0.01$ to $p = 1 - \frac{1}{H} = 0.005$. If we stop the reposi-

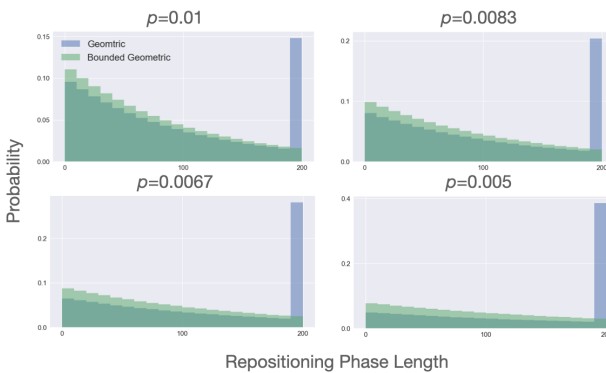

*Figure 4.* Distribution of length of repositioning phase (green) Bounded geometric distribution (blue) Original geometric distribution.

tioning phase with the probability $p = 1 - \gamma$ initially or equivalently sample the length from a geometric distribution with parameter $p$, a large amount of probability will accumulate at $H$. As shown in Figure 4 (blue), if the repositioning phase is stopped with probability $p$, there is a very large probability that it will consume one whole episode from the beginning of the training. As the training proceeds, the truncation probability decays, and this probability keeps increasing. This phenomenon will cause **Hyper** to be not as sample efficient in some tasks with a short horizon and requiring efficient exploration.

To address this issue, we use a bounded geometric distribution to sample the reposition length before every episode. The original geometric distribution probability function is defined on all positive integers, which is not desirable in this case. As in the episodic RL setting, the probability of values larger than $H$ accumulates to the marginal density of $H$, causing the undesirable spike shown in Figure 4. We instead restrict the domain to the range $[1, H]$ to avoid the probability accumulation, formulated as below:

$$\mathbb{P}(L = l) = \begin{cases} \frac{p(1-p)^{l-1}}{\sum_{i=1}^{H} p(1-p)^{(i-1)}} & 0 \leq l \leq H \\ 0 & \text{otherwise} \end{cases}$$

Figure 4 (green) depicts the shape of the resulting distribution for $p$ at different stages of the training. Despite there being an increasingly large probability for **Hyper** to have a longer repositioning phase, it still has sufficient opportunity to explore, which enhances **Hyper**'s capability of balancing exploration & exploitation in different cases.

## 6. Experiments

We evaluate the performance and the robustness of **Hyper** in this section, with the comparison to baselines with different strategies of balancing exploration & exploitation. We implement all algorithms with TD3 (Fujimoto et al.,

2018) as the reinforcement learning algorithm and with Disagreement (Pathak et al., 2019) as the intrinsic reward when curiosity-driven exploration is used. Specifically, we consider the following baselines:

**TD3 (Fujimoto et al., 2018)** Off-policy reinforcement learning method that uses random action noise for exploration.

**Curiosity-Driven Exploration (Curiosity)** Curiosity-driven TD3 agent, using Disagreement intrinsic reward for exploration.

**Decoupled Reinforcement Learning (Schäfer et al., 2021) (Decouple)** Agent that has an additional exploitation policy for exploiting the task by learning from the exploratory data collected by the exploration policy offline, as outlined in the previous section.

All baselines are evaluated on various environments that differ in exploration difficulty, exploitation difficulty, and function approximation difficulty. The detailed setting of the environments is deferred to the appendix. Decouple, Curiosity and Hyper all use the Disagreement method (Pathak et al., 2019) to compute the curiosity bonus for fair comparison.

### 6.1. Performance

Figure 5 depicts the performance of agents in the continuous goal-searching tasks (Fu et al., 2020) and locomotion tasks (Todorov et al., 2012) averaged over five trials, the shaded area represents the empirical standard deviation. The full results of the performance comparison and environment setup are deferred to the appendix.

In the goal-searching tasks (Figure 5), the agent is spawned following some initial distribution and will receive zero rewards until finding the fixed goal location. Hence the goal of the agent in this series of tasks is to first explore the environment and find the goal location, and then learn to exploit the task by consistently revisiting it. Two mazes with differences in size are used in the experiment: MediumMaze and LargeMaze, where it takes an optimal policy taking approximately 150 steps to reach the goal location in MediumMaze and 250 steps in LargeMaze. We differentiate the difficulty of tasks in the horizon of every episode. A shorter horizon results in a lower tolerance for redundant steps, which requires the agent to find a sufficiently good policy to obtain positive feedback from the environment. The horizon for each task is shown in the Table 2. In this set of experiments, we do not tune $\beta$. We apply $\beta = 1.0$ for **Hyper**, Decouple, and Curiosity agents, to ensure a fair comparison.

TD3 explores inefficiently with random actions, which

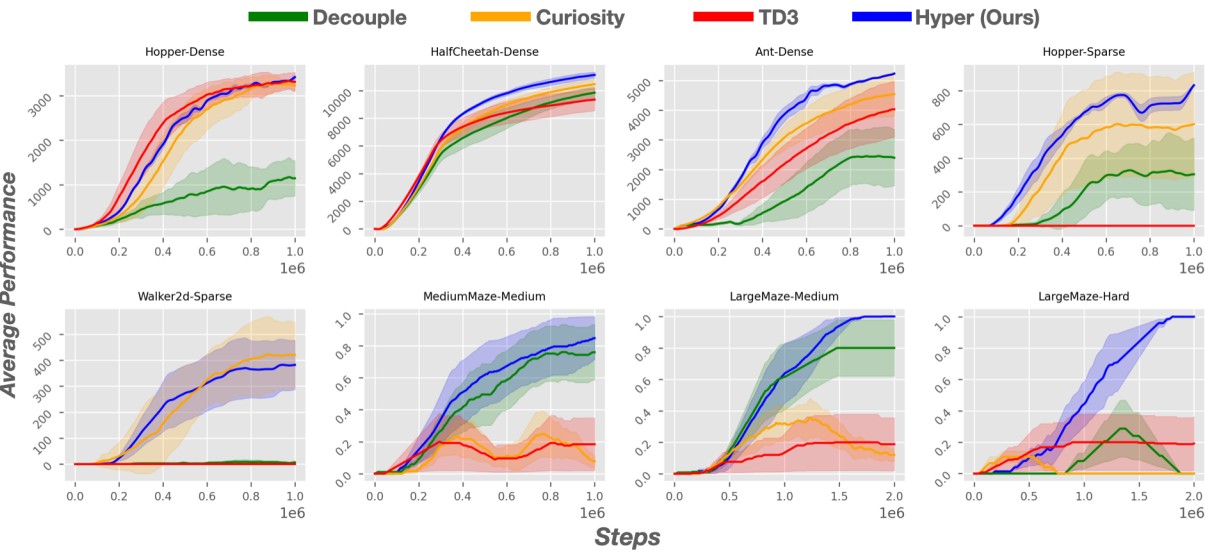

*Figure 5.* Performance of **Hyper** and baselines. For locomotion tasks, the performance is measured by the episodic cumulative reward, for navigation tasks, it is measured by the success rate instead. Each line is averaged over 5 runs with different random seeds.

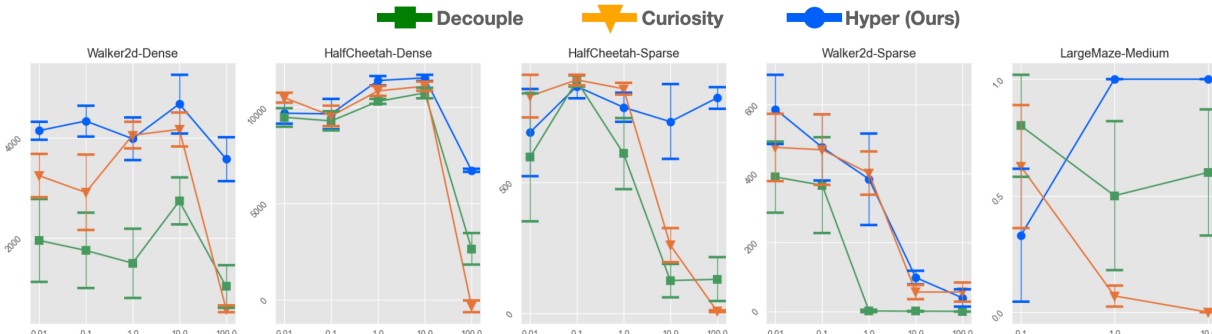

*Figure 6.* $\beta$-Sensitivity analysis. The x-axis represents the different choices of $\beta$, y-axis represents the final performance after 1M steps for the first four tasks, 2M steps for LargeMaze-Medium. The point in the graph represents the mean value over 5 runs, and the bars depict one standard deviation. The experiment shows that Hypernot only enjoys better robustness to $\beta$ compared to other methods, but also enjoys a smaller variance.

makes it hard to perform well in exploration-intensive tasks. In the sparse reward locomotion environment (Hopper-Sparse, Walker2d-Sparse), it fails to receive any reward in any trial. In the navigation tasks, despite it can quickly learn to exploit after finding the goal location, it can only manage to find the goal location once in all trials, indicating its lack of efficient exploration. Curiosity explores the environment efficiently and succeeds in the sparse-reward locomotion tasks. It also performs comparably, even favorably, in dense-reward locomotion tasks. However, the hard navigation tasks, show the problem in exploitation. It manages to find the goal location, however, it never manages to learn to properly exploit the task. The Decouple agent can learn a successful policy based on a limited number of success trajectories collected by the Curiosity agent in

MediumMaze-Medium and LargeMaze-Medium, but it fails to do so in the hardest navigation task, LargeMaze-Hard. The Decouple agent also struggles with high-dimensional locomotion tasks due to distribution shift.

**Hyper** performs comparable, often time favorable in all the tasks. Notably, **Hyper** is the only agent that consistently tackles the task. Compared to the failure of the Decouple agent in this task, it suggests that the regularization of the exploration visitation and the exploration persistence is crucial in this environment. The decouple agent fails in this task because it has no control over the exploration policy, and as the exploration policy cannot collect enough successful trajectories, it would be difficult for the Decouple agent to learn to exploit, whereas **Hyper** can guide the exploration policy towards the promising region, increasing the explo-

ration persistence, and the exploitation in turn benefits from the high-quality data.

## 6.2. $\beta$ Sensitivity Analysis

We now present the performance of **Hyper** and Curiosity with different intrinsic coefficients $\beta$.

We evaluate the final performance of Curiosity, Decouple, and **Hyper** agent over multiple trials with each different value of $\beta$ choice on five environments. As shown in Figure 6, the Curiosity agent shows peak performance with different values of $\beta$. Typically, the performance of the Curiosity agent peaks with a small value of $\beta$ and drastically drops as $\beta > 10$, whereas **Hyper** shows considerably more consistent with different $\beta$ values, indicating its robustness to this hyperparameter. It is worth noting that, **Hyper** generally shows better tolerance to the large value of $\beta$, which allows one to apply large $\beta$ with **Hyper** without prior knowledge of the environment, to exhibit more exploratory behavior and prevent from being stuck in sub-optimality.

## 7. Related Work

Exploration & exploitation is a long-standing research topic in the RL community (Thompson, 1933; Auer, 2002). Balancing exploration & exploitation is the key to efficient RL. Curiosity-driven exploration has emerged as a promising paradigm for achieving efficient RL. A line of theoretical research focuses on the optimal exploration-exploitation trade-off in RL with theoretically sound intrinsic rewards (Azar et al., 2017; Jin et al., 2018; Yang & Wang, 2020; Jin et al., 2020). The statistical properties of the environment (dynamics, rewards) and the intrinsic reward are assumed to be known, which allows one to find a coefficient $\beta$ for the optimal trade-off between exploration and exploitation.

Recently, this idea has been applied to practical RL algorithms and achieved great success in solving hard-exploration problems (Bellemare et al., 2016; Ostrovski et al., 2017; Burda et al., 2018). The common approach is to employ neural networks to estimate a transition's uncertainty and use this approximate uncertainty as the intrinsic reward $b$ for encouraging exploration. Various notions of uncertainty were used: error on dynamics prediction (Oudeyer et al., 2007; Pathak et al., 2017; 2019), state visitation density (Bellemare et al., 2016; Ostrovski et al., 2017; Machado et al., 2020), entropy gain (Tang et al., 2017; Choshen et al., 2018; Burda et al., 2018), fix-target prediction (Burda et al., 2018), etc.

Go-Explore is a powerful paradigm for efficient exploration that is not sensitive to the hyperparameter (Ecoffet et al., 2019). The algorithm keeps track of the state visitation, returns to the least visited states, and then explores from there. This paradigm does not use curiosity to exhibit exploration

but uses random actions. However, this method requires the environment to be either deterministic or resettable or for the agent to have access to some domain knowledge, which is generally unrealistic.

Bayesian RL is also proposed to exhibit efficient exploration (Ghavamzadeh et al., 2015; Fortunato et al., 2017; Osband et al., 2016), which leverages the ideas of Bayesian inference to quantify the uncertainty and encourage exploration accordingly. Bootstrapped DQN, the most practical variant of this line of work, wisely combines the idea of Bayesian inference and the property of the neural network for the agent to exhibit diverse and exploratory behavior and significantly improve the performance of DQN on a handful of environments. However, it does not fundamentally improve the exploration capability from the naive $\epsilon$-greedy exploration, as it fails in exploration-intensive tasks.

Options-based methods (Sutton et al., 1999; Bacon et al., 2017; Dabney et al., 2020; Chen et al., 2022; Kim et al., 2023) are also proposed for tackling the exploration problem by temporally abstracting the actions, resulting in conceptually easier policy learning and more consistent exploration. This paradigm is closely related to **Hyper**, whereas options-based methods mostly train an additional high-level switch policy to decide action selection protocol. (Kim et al., 2023) proposes an algorithm (LESSON) that automatically selects from exploitation and exploration policy to take action at every step, which yields superior results compared to original exploration and other option-based methods. We also compare **Hyper** to LESSON, examining the efficiency of **Hyper**. The results are deferred to the appendix.

Recently, some dedicated methods have been for mitigating the hyperparameter sensitivity of the exploration methods (Liu et al., 2021; Whitney et al., 2021; Schäfer et al., 2021). Among these methods, (Schäfer et al., 2021) is the most related to our method, as it uses the decoupling method as we outlined in Section 4 to mitigate the problem. It is efficient in low-dimensional cases, but hard to scale to tasks with high-dimensional state and action spaces, as we thoroughly demonstrated in the experiment.

## 8. Conclusion

We propose **Hyper**, a novel algorithm that leverages two distinct policies, that improve the robustness of curiosity-driven exploration by regularizing the exploration visitation. We theoretically justify its sample efficiency, that **Hyper** explores the environment efficiently. We also empirically validate its performance and robustness compared to the exploitation algorithm, curiosity-driven exploration algorithm, and previous attempts at solving this problem.

## Impact Statement

This paper presents work whose goal is to advance the field of Machine Learning. There are many potential societal consequences of our work, none which we feel must be specifically highlighted here.

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

# A. Implementation & Experiment Details

## A.1. Implementation of Agents

We implement **Hyper**, Decouple, and Curiosity based on the official implementation of TD3 (Fujimoto et al., 2018) along with the original hyperparameters reported in the paper. For adapting the intrinsic reward method with TD3, we update the Disagreement (Pathak et al., 2019) intrinsic reward model for every environmental step using $25\%$ of data of each batch to prevent the intrinsic reward from shrinking too fast following (Burda et al., 2018). In the locomotion experiments, we set the truncation probability $p$ to be 0.01 initially, and decay to 0.001, as we discussed in Section 5.

## A.2. PointMaze

Our experiments on continuous navigation tasks are conducted in the PointMaze domain (Todorov et al., 2012; Fu et al., 2020). PointMaze mostly serves as a fully observable goal-reaching benchmark, where the agent observes a 6 values at a state: current x-axis position, current y-axis position, current x-axis speed, current y-axis speed, goal x-axis position and goal y-axis position. And the agent is allowed to take a 2-dimensional action, that controls the x-axis acceleration and y-axis acceleration respectively.

In our experiment, we turn it into a sparse-reward goal searching domain. In our experiment, the agent only observes its own position and state, but not the goal location. The agent will only receive positive reward once reaching the goal. In this series of tasks, the agents not only need to first find the goal location, but also to consistently reach the goal. Despite the environments consisting of low-dimensional state and action space, this series of experiments examines and distinguishes the capability of balancing exploration & exploitation of agents.

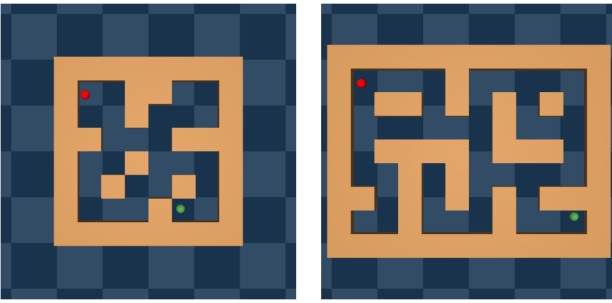

*Figure 7.* Layout of environments used continuous navigation experiments: (Left) MediumMaze (Right) LargeMaze

Figure 7 shows the layout of the MediumMaze and LargeMaze, where the initial location is marked with the green circle, and the goal location is marked with the red circle. For every episode, the agent is spawned randomly near the initial location, and the goal is spawned near the goal location. It takes approximately 150 steps for an optimal policy to reach the goal in MediumMaze and 250 steps in LargeMaze (depending on the randomly spawned agent and goal location).

We differentiate the difficulty of each task by restricting the horizon of each task. Specifically, tasks with shorter show less tolerance to redundant steps, and the agent will only receive positive feedback by exploring a more optimal trajectory. The detailed setup is shown in Table 2, where *Optimal #Steps* means the number of steps that an optimal policy needs to take from the initial location to the goal location, the variation is caused by the randomness when initializing the initial and goal location.

| Task Name | Optimal #Steps | Horizon $H$ |
|---|---|---|
| MediumMaze-Easy | ≈150 | 500 |
| MediumMaze-Medium | ≈150 | 200 |
| LargeMaze-Easy | ≈250 | 1000 |
| LargeMaze-Medium | ≈250 | 500 |
| LargeMaze-Hard | ≈250 | 300 |

*Table 2.* Difficulty of navigation tasks

### A.3. MuJoCo Locomotion

For the locomotion environments (Todorov et al., 2012), the agent starts idle and the task is to control the robot to move forward as fast as possible within 1000 steps, and the episode will end if the robot falls down. The agent will observe the position, velocity, and angular velocity of the joints of the robot, and take actions to control the torque on all the joints. In the dense reward version, including Hopper, Walker2d, HalfCheetah, Ant, and Humanoid. At each step, the agent will receive a performance reward proportional to its velocity and a constant "healthy reward" if it remains in a healthy position (i.e. not falling). In the sparse reward version, including SparseHopper, SparseWalker2d, SparseHalfCheetah, and SparseHumanoid, the agent will not receive the healthy reward, and will only get a unit reward once its forward speed exceeds some threshold.

### A.4. More Experiment Results

We present the full set of experiment results in this section. TD3 and Curiosity agent perform well in some of the tasks, depending on the property of the environment, whereas **Hyper** performs comparable to the best-performing algorithm in every task.

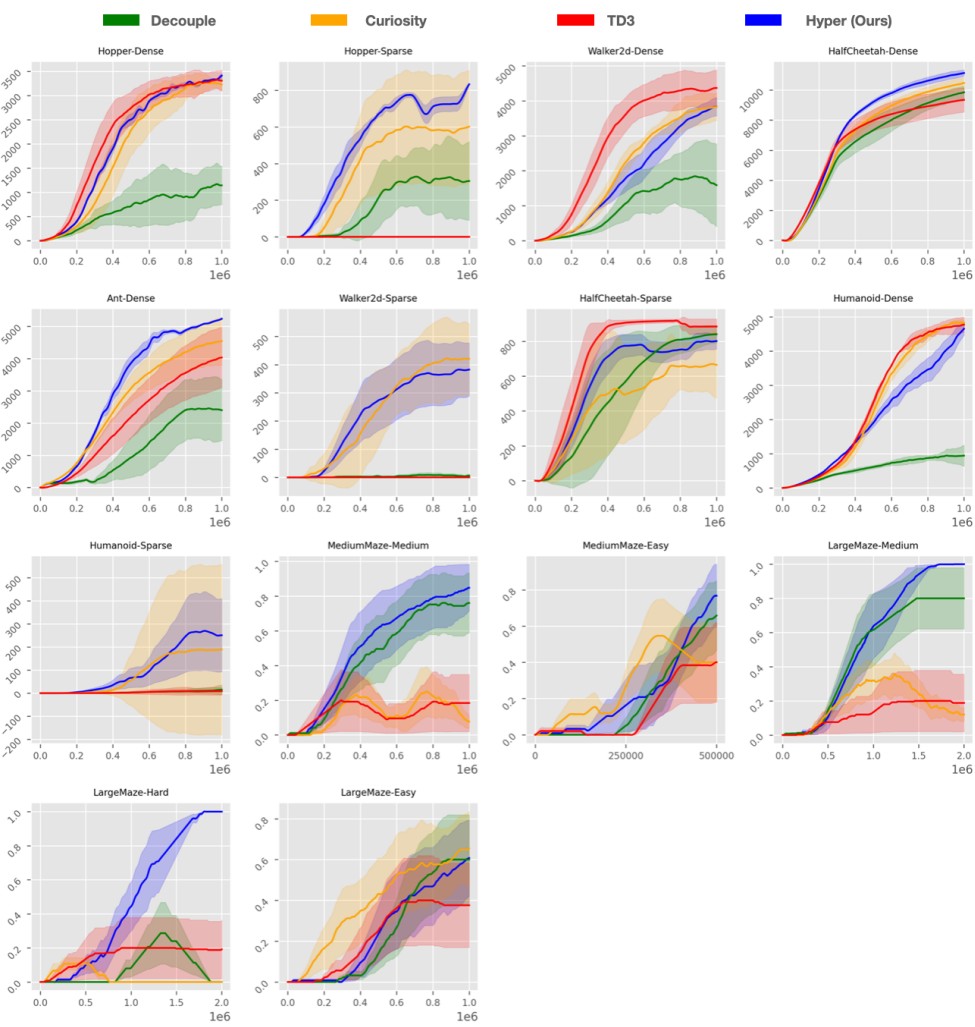

*Figure 8.* Full experiments results for performance comparison.

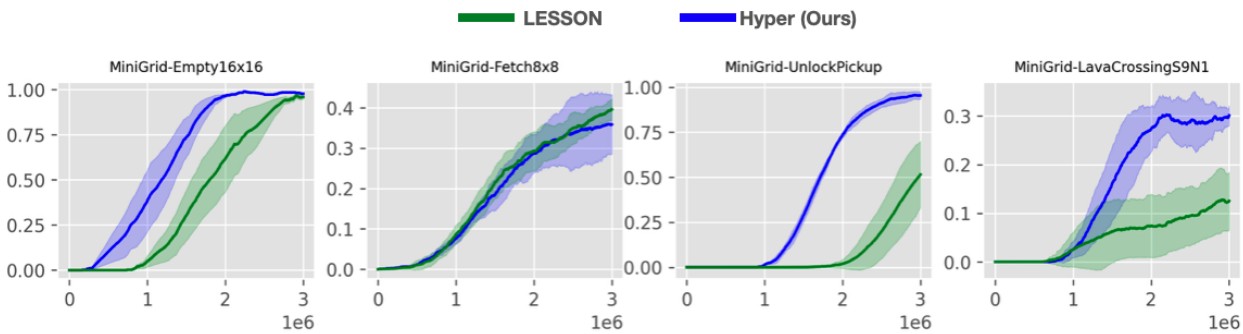

*Figure 9.* MiniGrid Performance of **Hyper** and LESSON. The x-axis represents the number of steps, y-axis represents the success rate.

### A.5. Comparision with Option-Based Method

We compare our method with the recently proposed options-based exploration method LESSON (Kim et al., 2023). We implement **Hyper** based on official implementation provided by (Kim et al., 2023), which uses DQN (Mnih et al., 2013) as the RL algorithm, and RND (Burda et al., 2018) as the intrinsic reward. We also use the same set of hyperparameters provided by the original paper.

We apply **Hyper** in the MiniGrid domain, on which LESSON is tested in the original paper. The comparison is shown in the Figure 9. **Hyper** performs comparable in Empty16x16 and Fetch8x8 tasks, and outperforms LESSON in UnlockPickup and LavaCrossingS9N1, which further validates the efficiency of **Hyper**.

### A.6. Hyperparameters of Experiments

#### A.6.1. HYPERPARAMETERS FOR TD3-BASED ALGORITHMS

| Hyperparameter | Value |
|---|---|
| Learning Rate | 3e-4 |
| Intrinsic Reward Learning Rate | 1e-4 |
| Batch Size | 256 |
| Policy Update Delay | 2 |
| Optimizer | Adam |
| $Q$-Network Architecture | (256, 256) |
| Actor-Network Architecture | (256, 256) |
| Activation function | ReLU |

#### A.6.2. HYPERPARAMETERS FOR CURIOSITY-DRIVEN EXPLORATION

| Hyperparameter | Value |
|---|---|
| $\beta$ | 1.0 |
| $p$ | (0.01, 0.001) |
| Learning Rate of Disagreement Model | 1e-4 |
| Disagreement Ensemble Size | 5 |

## B. Proof of Efficiency of Hyper

**Theorem B.1.** *With any truncation probability $p \in (0, 1)$ for the repositioning phase it takes at most $\tilde{\mathcal{O}}(\frac{d^3 H^4}{\epsilon^2})$ steps for Linear-UCB-**Hyper** to obtain an $\epsilon$-optimal exploitation policy $\mu$ with high probability under assumption B.2.*

In this section, we provide comprehensive proof for theoretically justifying the efficiency of Hyper under linear function approximation assumption (Linear-UCB-**Hyper**), which is formally stated in Theorem B.1. To theoretically justify the efficiency of **Hyper**, the core assumption is the following:

**Algorithm 2** Provably Efficient Linear-UCB-Hyper

---

**Requires:** Parameters $\lambda > 0, \mu > 0, \beta > 0, \beta' > 0$, Horizon $H$, Feature Mapping $\phi$, Truncation probability $p > 0$, Weights $\hat{\mathbf{w}}_h$ for optimistic $Q$-function, $Q$-function $\mathbf{w}_h$ for exploitation, $\check{\mathbf{w}}_h$ for pessimistic $Q$-function for all $h \in [H]$

**Requires:** Clipping function $\mathrm{clip}(x) : x \to \begin{cases} 0 & x \leq 0 \\ x & x \in (0, H) \\ H & x \geq H \end{cases}$, Geometrical distribution Geom

**for** $k = 1, 2, ..., K$ **do**
   Receive the initial state $s_1^k$
   Sample length $L^k \sim \mathrm{Geom}(p)$ for the first phase
   # Repositioning phase
   **for** step $h = 1, 2, ..., L^k$ **do**
      Take action $a_h^k \leftarrow arg \max_{a \in \mathcal{A}} Q_h(s_h^k, a; \mathbf{w}_h)$, and observe $s_{h+1}^k$
   **end for**
   # Exploration phase
   **for** step $h = L^k + 1, L^k + 2, ..., H$ **do**
      Take action $a_h^k \leftarrow arg \max_{a \in \mathcal{A}} \hat{Q}_h(s_h^k, a; \hat{\mathbf{w}}_h)$, and observe $s_{h+1}^k$
   **end for**
   # Policy Improvement
   **for** $h = H, H - 1, ..., 1$ **do**
      **if** $L^k = 0$ **then**
         $\Lambda_h \leftarrow \Sigma_{\tau=1}^{k-1} \phi(s_h^\tau, a_h^\tau) \phi(s_h^\tau, a_h^\tau)^T + \lambda \cdot \mathbf{I}$
         $\mathbf{w}_h \leftarrow \Lambda_h^{-1} \Sigma_{\tau=1}^{k-1} \phi(s_h^\tau, a_h^\tau)[r_h(s_h^\tau, a_h^\tau) + \max_a Q_{h+1}(s_h^{\tau+1}, a; \mathbf{w}_h)]$
         $\hat{\mathbf{w}}_h \leftarrow \Lambda_h^{-1} \Sigma_{\tau=1}^{k-1} \phi(s_h^\tau, a_h^\tau)[r_h(s_h^\tau, a_h^\tau) + \max_a \hat{Q}_{h+1}(s_h^{\tau+1}, a; \hat{\mathbf{w}}_h)]$
         $\check{\mathbf{w}}_h \leftarrow \Lambda_h^{-1} \Sigma_{\tau=1}^{k-1} \phi(s_h^\tau, a_h^\tau)[r_h(s_h^\tau, a_h^\tau) + \max_a \check{Q}_{h+1}(s_h^{\tau+1}, a; \check{\mathbf{w}}_h)]$
         $Q_h(\cdot, \cdot; \mathbf{w}_h) \leftarrow \mathrm{clip}\left(\mathbf{w}_h^T \phi(\cdot, \cdot)\right)$
         $\hat{Q}_h(\cdot, \cdot; \hat{\mathbf{w}}_h) \leftarrow \mathrm{clip}\left(\hat{\mathbf{w}}_h^T \phi(\cdot, \cdot) + \beta[\phi(\cdot, \cdot)^T \Lambda_h^{-1} \phi(\cdot, \cdot)]^{\frac{1}{2}}\right)$
         $\check{Q}_h(\cdot, \cdot; \check{\mathbf{w}}_h) \leftarrow \mathrm{clip}\left(\check{\mathbf{w}}_h^T \phi(\cdot, \cdot) - \beta'[\phi(\cdot, \cdot)^T \Lambda_h^{-1} \phi(\cdot, \cdot)]^{\frac{1}{2}}\right)$
      **end if**
   **end for**
**end for**

---

**Assumption B.2.** (Linear MDP, e.g., (Yang & Wang, 2019; Jin et al., 2020)). MDP$(\mathcal{S}, \mathcal{A}, H, \mathbb{P}, r)$ is a linear MDP whose transition $\mathbb{P} := \{\mathbb{P}_h\}_{h=1}^H$ is not necessarily stationary. With a feature map $\phi : \mathcal{S} \times \mathcal{A} \to \mathbb{R}^d$, such that for for any $h \in [H]$, there exists d unknown measures $\mu_h = (\mu_h^{(1)}, \mu_h^{(2)}, \mu_h^{(3)}, ..., \mu_h^{(d)})$ over $\mathcal{S}$ and an unknown vector $\theta_h \in \mathbb{R}^d$, such that for any $(s, a) \in \mathcal{S} \times \mathcal{A}$ we have:

$$\mathbb{P}_h(\cdot|s, a) = \phi(s, a)^T \mu_h(\cdot) \quad and \quad r_h(s, a) = \phi(s, a)^T \theta_h \tag{2}$$

Without loss of generality, we also assume that $\|\phi(s, a)\| \leq 1$, and $\max\{\|\mu_h(\mathcal{S})\|, \|\theta_h\|\} \leq \sqrt{d}$ for all $(s, a, h) \in \mathcal{S} \times \mathcal{A} \times [H]$

Specifically, we adopt the UCB-enhanced least-square value-iteration (Jin et al., 2020), a theoretically well-studied off-policy RL algorithm to our proposed Hyper framework, which we refer to as UCB-Hyper. The algorithm follows the generic Hyper framework shown in Algorithm **??**, it first collects data in a two-phase manner, and updates the policies afterward. Note that Linear-UCB-Hyper adopts an additional pessimistic $Q$-function with weights $\check{\mathbf{w}}_{h=1}^H$, this $Q$-function is not used in Linear-UCB-Hyper, but serves as a tool for our proof. We put it in the Algorithm 2 just for the sake of clarity in terms of the definition and the update rule.

For simplicity, we consider $\gamma = 1.0$ without loss of generality, we denote $\mathbf{w}_h, \hat{\mathbf{w}}_h, \check{\mathbf{w}}_h$ at $k$-th episode as $\mathbf{w}_h^k, \hat{\mathbf{w}}_h^k, \hat{\mathbf{w}}_h^k$, and denote $Q_h(\cdot, \cdot; \mathbf{w}_h), \hat{Q}_h(\cdot, \cdot; \mathbf{w}_h), \check{Q}_h(\cdot, \cdot; \check{\mathbf{w}}_h)$ at $k$-th episode as $Q_h^k(\cdot, \cdot), \hat{Q}_h^k(\cdot, \cdot), \check{Q}_h^k(\cdot, \cdot)$ when the context is clear.

**Proposition B.3.** *Bellman equation: (Bellman, 1957)*

$$Q_h^\pi(s, a) = r_h(s, a) + \gamma \mathbb{P}_h V_{h+1}^\pi(s, a) \quad and \quad V_h^\pi(s) = Q_h^\pi(s, \pi_h(s)), \quad \forall(s, a) \in \mathcal{S} \times \mathcal{A} \tag{3}$$

$$Q_h^\star(s, a) = r_h(s, a) + \gamma \mathbb{P}_h V_{h+1}^\star(s, a) \quad and \quad V_h^\star(s) = Q_h^\star(s, \pi_h^\star(s)), \quad \forall(s, a) \in \mathcal{S} \times \mathcal{A} \tag{4}$$

**Proposition B.4.** *($Q^\pi$ realizability (Jin et al., 2020)) For a linear MDP, for any policy $\pi$, there exist weights $\{\mathbf{w}^\pi\}_{h\in[H]}$ such that for any $(s, a, h) \in \mathcal{S} \times \mathcal{A} \times [H]$, we have $Q_h^\pi(s, a) = \phi(s, a)^T \mathbf{w}_h^\pi$.*

*Proof.* By the Bellman equation we have:

$$Q_h^\pi(s, a) = r(s, a) + (\mathbb{P}_h V_{h+1}^\pi)(s, a) = \phi(s, a)^T \theta_h + \int_{\mathcal{S}} V_{h+1}^\pi(s') \cdot \phi(s, a)^T d\mu_h(s')$$

$$= \phi(s, a)^T \cdot \left(\theta_h + \int_{\mathcal{S}} V_{h+1}^\pi(s') d\mu_h(s')\right). \tag{5}$$

This directly shows that $Q_h^\pi$ is linear with respect to features $\phi$. $\square$

**Lemma B.5.** *(Boundedness of $w_h^\pi$ (Jin et al., 2020)) Under Assumption B.2 for any fixed policy $\pi$, let $\{\mathbf{w}_h^\pi\}_{h\in[H]}$ be the weights such that $Q_h^\pi(s, a) = \langle \phi(s, a), \mathbf{w}_h^\pi \rangle$ for all $(s, a, h) \in \mathcal{S} \times \mathcal{A} \times [H]$. Then, we have*

$$\|\mathbf{w}_h^\pi\| \le 2H\sqrt{d}, \quad \forall h \in [H]$$

*Proof. By the Bellman equation, we have:*

$$Q_h^\pi(s, a) = \left(r_h + \mathbb{P}_h V_{h+1}^\pi\right)(s, a), \quad \forall h \in [H]$$

*And by the Proposition B.4, we have:*

$$\mathbf{w}_h^\pi = \theta_h + \int V_{h+1}^\pi(s') \, \mathrm{d}\mu_h(s')$$

*Under the normalization conditions of Assumption B.2 the reward at each step is in [0,1], we have:*

$$V_{h+1}^\pi(s') \le H, \quad \forall s' \sim \mathbb{P}(\cdot|s, a)$$

*Thus, $\|\theta_h\| \le \sqrt{d}$, and $\left\|\int V_{h+1}^\pi(s') \, \mathrm{d}\mu_h(s')\right\| \le H\sqrt{d}$. This concludes the proof.*

**Lemma B.6.** *(Bound on $\hat{\mathbf{w}}_h^k$ in Algorithm 2 (Jin et al., 2020)) The weight $\hat{\mathbf{w}}_h^k$ in Algorihtm 2 satisfies:*

$$\left\|\hat{\mathbf{w}}_h^k\right\| \le 2H\sqrt{dk/\lambda}$$

*Proof.* For simplicity, we denote the index set $\mathbf{U}^k = \{i \in [K] : L^i = 0\}$, i.e. the index of episodes in which the roll-in length is 0. For any index of episode $k \in [K]$, we denote $\lfloor k \rfloor = \max(\mathbf{U}^k)$ when $U$ is not empty, and $\lfloor k \rfloor = 0$ otherwise, i.e. the last time we encounter an episode whose roll-in length is 0. Suppose $\mathbf{v} \in \mathbb{R}^d$ is an arbitrary vector, we have:

$$\left|\mathbf{v}^\top \hat{\mathbf{w}}_h^k\right| = \left|\mathbf{v}^\top \left(\Lambda_h^{\lfloor k \rfloor}\right)^{-1} \sum_{\tau=1}^{\lfloor k \rfloor - 1} \phi_h^\tau \left[r(s_h^\tau, a_h^\tau) + \max_a \hat{Q}_{h+1}\left(s_{h+1}^\tau, a\right)\right]\right| \tag{6}$$

$$\le \sum_{\tau=1}^{\lfloor k \rfloor - 1} \left|\mathbf{v}^\top \left(\Lambda_h^{\lfloor k \rfloor}\right)^{-1} \phi_h^\tau\right| \cdot 2H \tag{7}$$

$$\le \sqrt{\left[\sum_{\tau=1}^{\lfloor k \rfloor - 1} \mathbf{v}^\top \left(\Lambda_h^{\lfloor k \rfloor}\right)^{-1} \mathbf{v}\right] \cdot \left[\sum_{\tau=1}^{\lfloor k \rfloor - 1} (\phi_h^\tau)^\top \left(\Lambda_h^{\lfloor k \rfloor}\right)^{-1} \phi_h^\tau\right]} \cdot 2H \tag{8}$$

$$\le 2H\|\mathbf{v}\|\sqrt{d\lfloor k \rfloor/\lambda} \tag{9}$$

where the first step follows the algorithm construction, the second step follows directly from Cauchy–Schwarz inequality, and the last step follows from Lemma B.1, and the third step follows from the fact that $\left\|\hat{\mathbf{w}}_h^k\right\| = \max_{\mathbf{v}:\|\mathbf{v}\|=1} \left|\mathbf{v}^\top \hat{\mathbf{w}}_h^k\right|$.

This implies that $\|\hat{\mathbf{w}}_h^k\| \leq 2H\sqrt{d\lfloor k\rfloor/\lambda}$, and by definition of $\lfloor k\rfloor$, we have $\|\hat{\mathbf{w}}_h^k\| \leq 2H\sqrt{d\lfloor k\rfloor/\lambda} \leq 2H\sqrt{dk/\lambda}$, which concludes the proof.

$\square$

*Remark* B.7. Let $\beta' = c' \cdot dH\sqrt{log(2dT/\delta)}$ for some proper constant $c' > 0$, $\check{\mathbf{w}}_h^k \leftarrow \Lambda_h^{-1}\Sigma_{\tau=1}^{k-1}\phi(s_h^\tau, a_h^\tau)[r_h(s_h^\tau, a_h^\tau) + \max_a \check{Q}_{h+1}(s_h^{\tau+1}, a)]$, and $\check{Q}_h^k(\cdot, \cdot) \leftarrow \text{clip}\left(\{\check{\mathbf{w}}_h^T\phi(\cdot, \cdot) - \beta'[\phi(\cdot, \cdot)^T\Lambda_h^{-1}\phi(\cdot, \cdot)]^{\frac{1}{2}}\}, 0, H\right)$. By similar approach as in the proof of Lemma B.6, the weight $\check{\mathbf{w}}_h^k$ also satisfies:

$$\left\|\check{\mathbf{w}}_h^k\right\| \leq 2H\sqrt{dk/\lambda}$$

This result is direct, as the proof of Lemma B.6 does not leverage any property specific to $\hat{\mathbf{w}}_h^k$.

We then define a high-probability event that bound the approximation error of our optimistic value function.

**Lemma B.8.** *(High Probability Event on Approximating Optimistic Value Function (Jin et al., 2020)) Under the setting of Theorem B.1, let $c_\beta$ be the constant in the definition of $\beta$, such that*

$$\beta = c_\beta \cdot dH\sqrt{log(2dT/\delta)}.$$

*There exists and an absolute constant $C$ that is independent of $c_\beta$ such that for any fixed $p \in [0, 1]$, if we let $\mathcal{E}$ be the event that:*

$$\forall (k, h) \in [K] \times [H] : \quad \left\|\sum_{\tau=1}^{k-1}\phi_h^\tau\left[\hat{V}_{h+1}^k\left(s_{h+1}^\tau\right) - \mathbb{P}_h\hat{V}_{h+1}^k\left(s_h^\tau, a_h^\tau\right)\right]\right\|_{\left(\Lambda_h^k\right)^{-1}} \leq C \cdot dH\sqrt{\chi}$$

*where $\chi = \log\left[2\left(c_\beta + 1\right)dT/p\right]$, then $\mathbb{P}(\mathcal{E}) \geq 1 - p/2$.*

*Proof.* By Lemma B.6, we have:
$$\left\|\hat{\mathbf{w}}_h^k\right\| \leq 2H\sqrt{dk/\lambda}, \quad \forall (k, h) \in [K] \times [H]$$

Also, by the construction of $\Lambda_h^k$, its smallest eigenvalue is lower bounded by $\lambda$. Combining with Lemmas C.5 and C.7, for any fixed constant $\epsilon > 0$, we have:

$$\left\|\sum_{\tau=1}^{k-1}\phi_h^\tau\left[\hat{V}_{h+1}^k\left(s_{h+1}^\tau\right) - \mathbb{P}_h\hat{V}_{h+1}^k\left(s_h^\tau, a_h^\tau\right)\right]\right\|_{\left(\Lambda_h^k\right)^{-1}}^2 \tag{10}$$

$$\leq 4H^2\left[\frac{d}{2}\log\left(\frac{k+\lambda}{\lambda}\right) + d\log\left(1 + \frac{8H\sqrt{dk}}{\varepsilon\sqrt{\lambda}}\right) + d^2\log\left(1 + \frac{8d^{1/2}\beta^2}{\varepsilon^2\lambda}\right) + \log\left(\frac{2}{p}\right)\right] + \frac{8k^2\varepsilon^2}{\lambda} \tag{11}$$

By plugging in $\lambda = 1$ and $\beta = C \cdot dH\sqrt{log(2dT/\delta)}$ to this inequality, where $C$ is a positive constant independent of $c_\beta$, and picking $\epsilon = dH/k$ we have:

$$\left\|\sum_{\tau=1}^{k-1}\phi_h^\tau\left[\hat{V}_{h+1}^k\left(s_{h+1}^\tau\right) - \mathbb{P}_h\hat{V}_{h+1}^k\left(s_h^\tau, a_h^\tau\right)\right]\right\|_{\left(\Lambda_h^k\right)^{-1}}^2 \leq C \cdot d^2H^2\log\left[2\left(c_\beta + 1\right)dT/p\right],$$

This concludes the proof.

$\square$

Lemma B.8 provides the bound on the approximation of the optimistic value function, we can then bound the pessimistic value function in a similar way, as by Lemma C.8, these two functions classes share the same upper bound on the covering number.

**Lemma B.9.** *(High Probability Event on Approximating Pessimistic Value Function) Under the setting of Theorem B.1, let $c_{\beta'}$ be the constant in the definition of $\beta'$, such that*

$$\beta' = c_{\beta'} \cdot dH\sqrt{log(2dT/\delta)}.$$

*There exists and an absolute constant $C'$ that is independent of $c_{\beta'}$ such that for any fixed $p \in [0, 1]$, if we let $\mathcal{E}$ be the event that:*

$$\forall (k, h) \in [K] \times [H]: \quad \left\| \sum_{\tau=1}^{k-1} \phi_h^\tau \left[ \check{V}_{h+1}^k \left(s_{h+1}^\tau\right) - \mathbb{P}_h \check{V}_{h+1}^k \left(s_h^\tau, a_h^\tau\right) \right] \right\|_{\left(\Lambda_h^k\right)^{-1}} \leq C' \cdot dH\sqrt{\chi}$$

*where $\chi = \log\left[2\left(c_{\beta'}+1\right)dT/p\right]$, then $\mathbb{P}(\mathcal{E}) \geq 1 - p/2$.*

*Proof.* By Lemma B.6, we have:

$$\left\|\check{\mathbf{w}}_h^k\right\| \leq 2H\sqrt{dk/\lambda}, \quad \forall (k, h) \in [K] \times [H]$$

Also, by the construction of $\Lambda_h^k$, its smallest eigenvalue is lower bounded by $\lambda$. Combining with Lemmas C.5 and C.8, for any fixed constant $\epsilon > 0$, we have:

$$\left\| \sum_{\tau=1}^{k-1} \phi_h^\tau \left[ \check{V}_{h+1}^k \left(s_{h+1}^\tau\right) - \mathbb{P}_h \check{V}_{h+1}^k \left(s_h^\tau, a_h^\tau\right) \right] \right\|_{\left(\Lambda_h^k\right)^{-1}}^2 \tag{12}$$

$$\leq 4H^2 \left[ \frac{d}{2} \log\left(\frac{k+\lambda}{\lambda}\right) + d\log\left(1 + \frac{8H\sqrt{dk}}{\varepsilon\sqrt{\lambda}}\right) + d^2 \log\left(1 + \frac{8d^{1/2}\beta^2}{\varepsilon^2\lambda}\right) + \log\left(\frac{2}{p}\right) \right] + \frac{8k^2\varepsilon^2}{\lambda} \tag{13}$$

By plugging in $\lambda = 1$ and $\beta' = C' \cdot dH\sqrt{log(2dT/\delta)}$ to this inequality, where $C'$ is a positive constant independent of $c_{\beta'}$, and picking $\epsilon = dH/k$ we have:

$$\left\| \sum_{\tau=1}^{k-1} \phi_h^\tau \left[ \check{V}_{h+1}^k \left(s_{h+1}^\tau\right) - \mathbb{P}_h \check{V}_{h+1}^k \left(s_h^\tau, a_h^\tau\right) \right] \right\|_{\left(\Lambda_h^k\right)^{-1}}^2 \leq C' \cdot d^2 H^2 \log\left[2\left(c_\beta+1\right)dT/p\right],$$

This concludes the proof. $\square$

**Lemma B.10.** *(Optimistic Policy Action-Value Estimation Error (Jin et al., 2020)) There exists an absolute constant $c_\beta$ such that for $\beta = c_\beta \cdot dH\sqrt{\log(2dT/\delta)}$, and for any fixed policy $\pi$, on the high-probability event $\mathcal{E}$ defined in Lemma B.8 we have for all $(s, a, h, k) \in \mathcal{S} \times \mathcal{A} \times [H] \times [K]$ that:*

$$\left\langle \phi(s, a), \hat{\mathbf{w}}_h^k \right\rangle - Q_h^\pi(s, a) = \mathbb{P}_h \left(\hat{V}_{h+1}^k - V_{h+1}^\pi\right)(s, a) + \Delta_h^k(s, a),$$

*for some $\Delta_h^k(s, a)$ that satisfies $\left|\Delta_h^k(s, a)\right| \leq \beta\sqrt{\phi(s, a)^\top \left(\Lambda_h^k\right)^{-1} \phi(s, a)}$.*

*Proof.* By Proposition B.4 and the Equation 3, we know for any $(s, a, h) \in \mathcal{S} \times \mathcal{A} \times [H]$:

$$Q_h^\pi(s, a) := \left\langle \phi(s, a), \mathbf{w}_h^\pi \right\rangle = \left(r_h + \mathbb{P}_h V_{h+1}^\pi\right)(s, a)$$

And the residual between $\hat{\mathbf{w}}_h^k, \mathbf{w}_h^\pi$ is given by and can be decomposed as the following:

$$\hat{\mathbf{w}}_h^k - \mathbf{w}_h^\pi = \left(\Lambda_h^k\right)^{-1} \sum_{\tau=1}^{k-1} \boldsymbol{\phi}_h^\tau \left[r_h^\tau + \hat{V}_{h+1}^k \left(s_{h+1}^\tau\right)\right] - \mathbf{w}_h^\pi \tag{14}$$

$$= \left(\Lambda_h^k\right)^{-1} \left\{ -\lambda \mathbf{w}_h^\pi + \sum_{\tau=1}^{k-1} \boldsymbol{\phi}_h^\tau \left[\hat{V}_{h+1}^k \left(s_{h+1}^\tau\right) - \mathbb{P}_h V_{h+1}^\pi \left(s_h^\tau, a_h^\tau\right)\right] \right\} \tag{15}$$

$$= \underbrace{-\lambda \left(\Lambda_h^k\right)^{-1} \mathbf{w}_h^\pi}_{\mathbf{q}_1} + \underbrace{\left(\Lambda_h^k\right)^{-1} \sum_{\tau=1}^{k-1} \boldsymbol{\phi}_h^\tau \left[\hat{V}_{h+1}^k \left(s_{h+1}^\tau\right) - \mathbb{P}_h \hat{V}_{h+1}^k \left(s_h^\tau, a_h^\tau\right)\right]}_{\mathbf{q}_2} \tag{16}$$

$$+ \underbrace{\left(\Lambda_h^k\right)^{-1} \sum_{\tau=1}^{k-1} \boldsymbol{\phi}_h^\tau \mathbb{P}_h \left(\hat{V}_{h+1}^k - V_{h+1}^\pi\right) \left(s_h^\tau, a_h^\tau\right)}_{\mathbf{q}_3}. \tag{17}$$

Now, we bound the terms on the right-hand side individually. For the first term,

$$|\langle \boldsymbol{\phi}(s,a), \mathbf{q}_1 \rangle| = \left| \lambda \left\langle \boldsymbol{\phi}(s,a), \left(\Lambda_h^k\right)^{-1} \mathbf{w}_h^\pi \right\rangle \right| \le \sqrt{\lambda} \|\mathbf{w}_h^\pi\| \sqrt{\boldsymbol{\phi}(s,a)^\top \left(\Lambda_h^k\right)^{-1} \boldsymbol{\phi}(s,a)}.$$

For the second term, given the event $\mathcal{E}$ defined in Lemma B.8, we have:

$$|\langle \boldsymbol{\phi}(s,a), \mathbf{q}_2 \rangle| \le c_0 \cdot dH\sqrt{\chi} \sqrt{\boldsymbol{\phi}(s,a)^\top \left(\Lambda_h^k\right)^{-1} \boldsymbol{\phi}(s,a)}$$

for an absolute constant $c_0$ independent of $c_\beta$, and $\chi = \log\left[2\left(c_\beta + 1\right) dT/p\right]$. For the third term,

$$\langle \boldsymbol{\phi}(s,a), \mathbf{q}_3 \rangle = \left\langle \boldsymbol{\phi}(s,a), \left(\Lambda_h^k\right)^{-1} \sum_{\tau=1}^{k-1} \boldsymbol{\phi}_h^\tau \mathbb{P}_h \left(\hat{V}_{h+1}^k - V_{h+1}^\pi\right) \left(x_h^\tau, a_h^\tau\right) \right\rangle \tag{18}$$

$$= \left\langle \boldsymbol{\phi}(s,a), \left(\Lambda_h^k\right)^{-1} \sum_{\tau=1}^{k-1} \boldsymbol{\phi}_h^\tau \left(\boldsymbol{\phi}_h^\tau\right)^\top \int \left(\hat{V}_{h+1}^k - V_{h+1}^\pi\right) \left(x'\right) \mathrm{d}\boldsymbol{\mu}_h \left(x'\right) \right\rangle \tag{19}$$

$$= \underbrace{\left\langle \boldsymbol{\phi}(s,a), \int \left(\hat{V}_{h+1}^k - V_{h+1}^\pi\right) \left(x'\right) \mathrm{d}\boldsymbol{\mu}_h \left(x'\right) \right\rangle}_{p_1} \underbrace{-\lambda \left\langle \boldsymbol{\phi}(s,a), \left(\Lambda_h^k\right)^{-1} \int \left(\hat{V}_{h+1}^k - V_{h+1}^\pi\right) \left(x'\right) \mathrm{d}\boldsymbol{\mu}_h \left(x'\right) \right\rangle}_{p_2}, \tag{20}$$

where, by Assumption B.2 Equation 1, we have

$$p_1 = \mathbb{P}_h \left(\hat{V}_{h+1}^k - V_{h+1}^\pi\right) (s,a), \quad |p_2| \le 2H\sqrt{d\lambda} \sqrt{\boldsymbol{\phi}(s,a)^\top \left(\Lambda_h^k\right)^{-1} \boldsymbol{\phi}(s,a)}$$

Finally, since $\langle \boldsymbol{\phi}(s,a), \hat{\mathbf{w}}_h^k \rangle - Q_h^\pi(s,a) = \langle \boldsymbol{\phi}(s,a), \hat{\mathbf{w}}_h^k - \mathbf{w}_h^\pi \rangle = \langle \boldsymbol{\phi}(s,a), \mathbf{q}_1 + \mathbf{q}_2 + \mathbf{q}_3 \rangle$, by Lemma B.5 and our choice of parameter $\lambda$, we have

$$\left| \langle \boldsymbol{\phi}(s,a), \hat{\mathbf{w}}_h^k \rangle - Q_h^\pi(s,a) - \mathbb{P}_h \left(\hat{V}_{h+1}^k - V_{h+1}^\pi\right) (s,a) \right| \le c' \cdot dH\sqrt{\chi} \sqrt{\boldsymbol{\phi}(s,a)^\top \left(\Lambda_h^k\right)^{-1} \boldsymbol{\phi}(s,a)},$$

for an absolute constant $c'$ independent of $c_\beta$. Finally, to prove this lemma, we only need to show that there exists a choice of absolute constant $c_\beta$ so that

$$c'\sqrt{\iota + \log\left(c_\beta + 1\right)} \le c_\beta \sqrt{\iota} \tag{21}$$

where $\iota = \log(2dT/p)$. We know $\iota \in [\log 2, \infty)$ by its definition, and $c'$ is an absolute constant independent of $c_\beta$. Therefore, we can pick an absolute constant $c_\beta$ which satisfies $c'\sqrt{\log 2 + \log\left(c_\beta + 1\right)} \le c_\beta \sqrt{\log 2}$. This choice of $c_\beta$ will make Equation 21 hold for all $\iota \in [\log 2, \infty)$, which finishes the proof.

$\square$

With similar approach, we can bound the action-value approximation for the pessimistic policy.

**Lemma B.11.** *(Pessimistic Policy Action-Value Estimation Error) There exists an absolute constant $c_{\beta'}$ such that for $\beta' = c_{\beta'} \cdot dH\sqrt{\log(2dT/p)}$, and for any fixed policy $\pi$, on the high-probability event $\mathcal{E}$ defined in Lemma B.9 we have for all $(s, a, h, k) \in \mathcal{S} \times \mathcal{A} \times [H] \times [K]$ that:*

$$\left\langle \phi(s,a), \check{\mathbf{w}}_h^k \right\rangle - Q_h^\pi(s,a) = \mathbb{P}_h \left( \check{V}_{h+1}^k - V_{h+1}^\pi \right)(s,a) + \tilde{\Delta}_h^k(s,a),$$

*for some $\tilde{\Delta}_h^k(s,a)$ that satisfies $\left| \tilde{\Delta}_h^k(s,a) \right| \leq \beta' \sqrt{\phi(s,a)^\top \left( \Lambda_h^k \right)^{-1} \phi(s,a)}$.*

*Proof.* Similar to the proof of Lemma B.10, we decompose the residule between $\check{\mathbf{w}}_h^k$ and $\mathbf{w}_h^\pi$ as the following:

$$\check{\mathbf{w}}_h^k - \mathbf{w}_h^\pi = \left( \Lambda_h^k \right)^{-1} \sum_{\tau=1}^{k-1} \phi_h^\tau \left[ r_h^\tau + \check{V}_{h+1}^k \left( s_{h+1}^\tau \right) \right] - \mathbf{w}_h^\pi \tag{22}$$

$$= \left( \Lambda_h^k \right)^{-1} \left\{ -\lambda \mathbf{w}_h^\pi + \sum_{\tau=1}^{k-1} \phi_h^\tau \left[ \check{V}_{h+1}^k \left( s_{h+1}^\tau \right) - \mathbb{P}_h V_{h+1}^\pi \left( s_h^\tau, a_h^\tau \right) \right] \right\} \tag{23}$$

$$= \underbrace{-\lambda \left( \Lambda_h^k \right)^{-1} \mathbf{w}_h^\pi}_{\mathbf{q}_1} + \underbrace{\left( \Lambda_h^k \right)^{-1} \sum_{\tau=1}^{k-1} \phi_h^\tau \left[ \check{V}_{h+1}^k \left( s_{h+1}^\tau \right) - \mathbb{P}_h \check{V}_{h+1}^k \left( s_h^\tau, a_h^\tau \right) \right]}_{\mathbf{q}_2} \tag{24}$$

$$+ \underbrace{\left( \Lambda_h^k \right)^{-1} \sum_{\tau=1}^{k-1} \phi_h^\tau \mathbb{P}_h \left( \check{V}_{h+1}^k - V_{h+1}^\pi \right) \left( s_h^\tau, a_h^\tau \right)}_{\mathbf{q}_3}. \tag{25}$$

By the proof of the first term,

$$\left| \left\langle \phi(s,a), \mathbf{q}_1 \right\rangle \right| = \left| \lambda \left\langle \phi(s,a), \left( \Lambda_h^k \right)^{-1} \mathbf{w}_h^\pi \right\rangle \right| \leq \sqrt{\lambda} \left\| \mathbf{w}_h^\pi \right\| \sqrt{\phi(s,a)^\top \left( \Lambda_h^k \right)^{-1} \phi(s,a)}.$$

For the second term, given the event $\mathcal{E}$ defined in Lemma B.9, we have:

$$\left| \left\langle \phi(s,a), \mathbf{q}_2 \right\rangle \right| \leq c_0 \cdot dH\sqrt{\chi} \sqrt{\phi(s,a)^\top \left( \Lambda_h^k \right)^{-1} \phi(s,a)}$$

for an absolute constant $c_0$ independent of $c_\beta$, and $\chi = \log \left[ 2 \left( c_{\beta'} + 1 \right) dT/p \right]$. For the third term,

$$\left\langle \phi(s,a), \mathbf{q}_3 \right\rangle = \left\langle \phi(s,a), \left( \Lambda_h^k \right)^{-1} \sum_{\tau=1}^{k-1} \phi_h^\tau \mathbb{P}_h \left( \check{V}_{h+1}^k - V_{h+1}^\pi \right) \left( x_h^\tau, a_h^\tau \right) \right\rangle \tag{26}$$

$$= \left\langle \phi(s,a), \left( \Lambda_h^k \right)^{-1} \sum_{\tau=1}^{k-1} \phi_h^\tau \left( \phi_h^\tau \right)^\top \int \left( \check{V}_{h+1}^k - V_{h+1}^\pi \right) (x') \, \mathrm{d}\boldsymbol{\mu}_h (x') \right\rangle \tag{27}$$

$$= \underbrace{\left\langle \phi(s,a), \int \left( \check{V}_{h+1}^k - V_{h+1}^\pi \right) (x') \, \mathrm{d}\boldsymbol{\mu}_h (x') \right\rangle}_{p_1} \tag{28}$$

$$\underbrace{-\lambda \left\langle \phi(s,a), \left( \Lambda_h^k \right)^{-1} \int \left( \check{V}_{h+1}^k - V_{h+1}^\pi \right) (x') \, \mathrm{d}\boldsymbol{\mu}_h (x') \right\rangle}_{p_2} \tag{29}$$

where, by Equation (3), we have

$$p_1 = \mathbb{P}_h \left( \check{V}_{h+1}^k - V_{h+1}^\pi \right)(s,a), \quad |p_2| \leq 2H\sqrt{d\lambda}\sqrt{\phi(s,a)^\top \left( \Lambda_h^k \right)^{-1} \phi(s,a)}$$

Finally, since $\langle \phi(s,a), \check{\mathbf{w}}_h^k \rangle - Q_h^\pi(s,a) = \langle \phi(s,a), \check{\mathbf{w}}_h^k - \mathbf{w}_h^\pi \rangle = \langle \phi(s,a), \mathbf{q}_1 + \mathbf{q}_2 + \mathbf{q}_3 \rangle$, by Lemma B.5 and our choice of parameter $\lambda$, we have

$$\left| \langle \phi(s,a), \check{\mathbf{w}}_h^k \rangle - Q_h^\pi(s,a) - \mathbb{P}_h \left( \check{V}_{h+1}^k - V_{h+1}^\pi \right)(s,a) \right| \leq c'' \cdot dH\sqrt{\chi}\sqrt{\phi(s,a)^\top \left( \Lambda_h^k \right)^{-1} \phi(s,a)},$$

for an absolute constant $c''$ independent of $c_{\beta'}$. Finally, to prove this lemma, we only need to show that there exists a choice of absolute constant $c_{\beta'}$ so that

$$c'' \sqrt{\iota + \log\left(c_{\beta'} + 1\right)} \leq c_{\beta'} \sqrt{\iota} \tag{30}$$

where $\iota = \log(2dT/p)$. We know $\iota \in [\log 2, \infty)$ by its definition, and $c'$ is an absolute constant independent of $c_{\beta'}$. Therefore, we can pick an absolute constant $c_\beta$ which satisfies $c' \sqrt{\log 2 + \log\left(c_{\beta'} + 1\right)} \leq c_{\beta'} \sqrt{\log 2}$. This choice of $c_{\beta'}$ will make Equation 30 hold for all $\iota \in [\log 2, \infty)$, which finishes the proof.

$\square$

**Lemma B.12.** *(Upper Confidence Bound (Jin et al., 2020)) Under the setting of Theorem B.1 on the event $\mathcal{E}$ defined in Lemma B.8 we have $\hat{Q}_h^k(s,a) \geq Q_h^\star(s,a)$ for all $(s,a,h,k) \in \mathcal{S} \times \mathcal{A} \times [H] \times [K]$.*

*Proof.* We prove this lemma by induction.

First, we prove the base case, at the last step $H$. The statement holds because $\hat{Q}_H^k(s,a) \geq Q_H^\star(s,a)$. Since the value function at $H+1$ step is zero, by Lemma B.10 we have:

$$\left| \langle \phi(s,a), \hat{\mathbf{w}}_H^k \rangle - Q_H^\star(s,a) \right| \leq \beta \sqrt{\phi(s,a)^\top \left( \Lambda_H^k \right)^{-1} \phi(s,a)}.$$

Therefore, we know:

$$Q_H^\star(s,a) \leq \min \left\{ \langle \phi(s,a), \hat{\mathbf{w}}_H^k \rangle + \beta \sqrt{\phi(s,a)^\top \left( \Lambda_H^k \right)^{-1} \phi(s,a)}, H \right\} = Q_H^k(s,a).$$

Now, suppose the statement holds true at step $h+1$ and consider step $h$. Again, by LemmaB.4, we have:

$$\left| \langle \phi(s,a), \hat{\mathbf{w}}_h^k \rangle - Q_h^\star(s,a) - \mathbb{P}_h \left( \hat{V}_{h+1}^k - V_{h+1}^\star \right)(s,a) \right| \leq \beta \sqrt{\phi(s,a)^\top \left( \Lambda_h^k \right)^{-1} \phi(s,a)}.$$

By the induction assumption that $\mathbb{P}_h \left( \hat{V}_{h+1}^k - V_{h+1}^\star \right)(s,a) \geq 0$, we have:

$$Q_h^\star(s,a) \leq \min \left\{ \langle \phi(s,a), \hat{\mathbf{w}}_h^k \rangle + \beta \sqrt{\phi(s,a)^\top \left( \Lambda_h^k \right)^{-1} \phi(s,a)}, H \right\} = \hat{Q}_h^k(s,a),$$

which concludes the proof. $\square$

We will also be needing the following lemma, for lower bounding the value of our output policy $arg\max_{a \in \mathcal{A}} Q(s, \cdot)$. The following lemma shows that the pessimistic value function always lower bounds any policy value function.

**Lemma B.13.** *(Lower Confidence Bound) Under the setting of Theorem B.1 on the event $\mathcal{E}$ defined in Lemma B.9 we have, for any policy $\pi$, $\check{Q}_h^k(s,a) \leq Q_h^\pi(s,a)$ for all $(s,a,h,k) \in \mathcal{S} \times \mathcal{A} \times [H] \times [K]$.*

*Proof.* We prove this lemma by induction similar to we just did in Lemma B.12.

Consider a fixed, arbitrary policy $\pi$, first, we prove the base case, at the last step $H$. The statement holds because $Q_H^\pi(s, a) \geq \check{Q}_H^k(s, a)$. Since the value function at $H + 1$ step is zero, by Lemma B.11 we have:

$$\left| \langle \phi(s, a), \check{\mathbf{w}}_H^k \rangle - Q_H^\pi(s, a) \right| \leq \beta' \sqrt{\phi(s, a)^\top \left( \Lambda_H^k \right)^{-1} \phi(s, a)}.$$

Therefore, we know:

$$Q_H^\pi(s, a) \geq \text{clip} \left( \langle \phi(s, a), \check{\mathbf{w}}_H^k \rangle - \beta' \sqrt{\phi(s, a)^\top \left( \Lambda_H^k \right)^{-1} \phi(s, a)} \right) = \check{Q}_H^k(s, a).$$

Now, suppose the statement holds true at step $h + 1$ and consider step $h$. Again, by Lemma B.11, we have:

$$\left| \langle \phi(s, a), \check{\mathbf{w}}_h^k \rangle - Q_h^\pi(s, a) - \mathbb{P}_h \left( \check{V}_{h+1}^k - V_{h+1}^\pi \right)(s, a) \right| \leq \beta' \sqrt{\phi(s, a)^\top \left( \Lambda_h^k \right)^{-1} \phi(s, a)}.$$

By the induction assumption that $\mathbb{P}_h \left( \check{V}_{h+1}^k - V_{h+1}^\pi \right)(s, a) \leq 0$, we have:

$$Q_h^\pi(s, a) \geq \text{clip} \left( \langle \phi(s, a), \check{\mathbf{w}}_h^k \rangle - \beta' \sqrt{\phi(s, a)^\top \left( \Lambda_h^k \right)^{-1} \phi(s, a)}, 0, H \right) = \check{Q}_h^k(s, a),$$

which concludes the proof. $\qquad\square$

**Theorem B.14.** *(Pseudo Regret Bound) Under Assumption B.2, for any fixed constant $\delta \in (0, 1)$, with proper choice of $c > 0$, and if we set $\lambda = 1$, $\beta = c \cdot dH \sqrt{\log(2dT/\delta)}$, then with probability at least $1 - \delta$, the regret of interest of algorithm 2, $\mathbb{E} \left[ \sum_{k=1}^K V_1^\star(s_1^k) - V_1^{\pi_k}(s_1^k) \right]$, is at most $\tilde{\mathcal{O}} \left( \sqrt{d^3 H^3 T} \right)$, where $p$ is the parameter of geometric distribution.*

*Proof.* For simplicity, we use the notation:

$$\hat{\pi}_h^k(s, \cdot) = \arg \max_{a \in \mathcal{A}} \hat{Q}_h^k(s, \cdot) \quad \pi_h^k(s, \cdot) = \arg \max_{a \in \mathcal{A}} Q_h^k(s, \cdot)$$

We also denote $\mathbb{I} = \{k \in [K], L^k = 0\}$, an index set of episodes in which the trajectory is fully exploratory, then we have,

$$\mathbb{E} \left[ \sum_{k=1}^K V_1^\star(s_1^k) - V_1^{\pi_k}(s_1^k) \right] \leq \mathbb{E} \left[ \sum_{k=1}^K \hat{V}_1^k(s_1^k) - \check{V}_1^k(s_1^k) \right] \tag{31}$$

$$= \mathbb{E} \left[ \sum_{k=1}^K \hat{V}_1^{\lfloor k \rfloor}(s_1^k) - \check{V}_1^{\lfloor k \rfloor}(s_1^k) \right] \tag{32}$$

$$= \frac{1}{p} \cdot \sum_{k \in \mathbb{I}} \hat{V}_1^k(s_1^k) - \check{V}_1^k(s_1^k) \tag{33}$$

where the first step is the direct result of Lemmas B.12 and B.13, the second and the third steps are due to the construction of our algorithm, where we do not update weights until a full exploratory episode happens, and the expected interval of such event happening is $\frac{1}{p}$. And further,

$$\sum_{k\in\mathbb{I}} \hat{V}_1^k(s_1^k) - \check{V}_1^k(s_1^k) = \sum_{k\in\mathbb{I}} \hat{Q}_1^k(s_1^k, a_1^k) - \check{Q}_1^k(s_1^k, a_1'^k) \tag{34}$$

$$\leq \sum_{k\in\mathbb{I}} \hat{Q}_1^k(s_1^k, a_1^k) - \check{Q}_1^k(s_1^k, a_1^k) \tag{35}$$

$$= \sum_{k\in\mathbb{I}} \left\{ \Delta_h^k(s_1^k, a_1^k) - \tilde{\Delta}_h^k(s_1^k, a_1^k) + \mathbb{E}\left[ \hat{V}_2^k(s_2^k) - \check{V}_2^k(s_2^k) | s_1^k, a_1^k \right] \right\} \tag{36}$$

$$\leq \sum_{k\in\mathbb{I}} \left\{ \beta\sqrt{\phi(s_1^k, a_1^k)^\top \left(\Lambda_h^k\right)^{-1} \phi(s_1^k, a_1^k)} + \beta'\sqrt{\phi(s_1^k, a_1^k)^\top \left(\Lambda_h^k\right)^{-1} \phi(s_1^k, a_1^k)} \right.$$
$$\left. + \mathbb{E}\left[ \hat{V}_2^k(s_2^k) - \check{V}_2^k(s_2^k) | s_1^k, a_1^k \right] \right\} \tag{37}$$

$$= \sum_{k\in\mathbb{I}} \left\{ \underbrace{\beta\sqrt{\phi(s_1^k, a_1^k)^\top \left(\Lambda_h^k\right)^{-1} \phi(s_1^k, a_1^k)}}_{b_1^k} + \underbrace{\beta'\sqrt{\phi(s_1^k, a_1^k)^\top \left(\Lambda_h^k\right)^{-1} \phi(s_1^k, a_1^k)}}_{b_1'^k} \right. \tag{38}$$

$$\left. + \underbrace{\mathbb{E}\left[ \hat{V}_2^k(s_2^k) - \check{V}_2^k(s_2^k) | s_1^k, a_1^k \right] - (\hat{V}_2^k(s_2^k) - \check{V}_2^k(s_2^k))}_{\zeta_2^k} + (\hat{V}_2^k(s_2^k) - \check{V}_2^k(s_2^k)) \right\} \tag{39}$$

$$= \sum_{k\in\mathbb{I}} \left[ \hat{V}_2^k(s_2^k) - \check{V}_2^k(s_2^k) + b_1^k + b_1'^k + \zeta_2^k \right] \tag{40}$$

where, $a \in arg\max_{a\in\mathcal{A}} \hat{Q}_1^k(s_1^k, \cdot)$ and $a' \in arg\max_{a'\in\mathcal{A}} \check{Q}_1^k(s_1^k, \cdot)$.

By recursively applying Equation. (34), we have,

$$\sum_{k\in\mathbb{I}} \hat{V}_1^k(s_1^k) - \check{V}_1^k(s_1^k) \leq \sum_{k\in\mathbb{I}}\sum_{h=1}^{H} b_h^k + \sum_{k\in\mathbb{I}}\sum_{h=1}^{H} b_h'^k + \sum_{k\in\mathbb{I}}\sum_{h=1}^{H} \zeta_h^k \tag{41}$$

$$\tag{42}$$

We now bound each terms, for the first term in Equation (41), by Lemma C.2 and C.3:

$$\sum_{k\in\mathbb{I}}\sum_{h=1}^{H} b_h^k = \sum_{k\in\mathbb{I}}\sum_{h=1}^{H} \beta\sqrt{\phi(s_h^k, a_h^k)^\top \left(\Lambda_h^k\right)^{-1} \phi(s_h^k, a_h^k)} \tag{43}$$

$$\leq \sum_{h=1}^{H} \sqrt{Kp} \cdot \left[ \sum_{k\in\mathbb{I}} \beta\sqrt{\phi(s_h^k, a_h^k)^\top \left(\Lambda_h^k\right)^{-1} \phi(s_h^k, a_h^k)} \right] \tag{44}$$

$$\leq \beta\sqrt{Kp} \sum_{h=1}^{H} \sqrt{2\log\left[\frac{\det(\Lambda_h^k)}{\det(\Lambda_h^1)}\right]} \tag{45}$$

$$\leq \beta\sqrt{Kp} \sum_{h=1}^{H} \sqrt{2d\log\left[\frac{\lambda + k}{\lambda}\right]} \tag{46}$$

$$\leq H\beta\iota\sqrt{2dKp} \tag{47}$$

where, the second step follows from Cauchy–Schwarz inequality, the third step follows from the Lemma C.2 and C.3, and the second last step follows from the fact that $\|\phi(\cdot,\cdot)\| \leq 1$, and thus $\|\Lambda_h^k\| \leq \lambda + k$. And following the same logic, we have, for the second term in Equation (41):

$$\sum_{k \in \mathbb{I}} \sum_{h=1}^{H} b_h'^k \leq H \beta' \iota \sqrt{2dKp}$$

For the third term in Eq(41), we notice it is a martingale difference sequence, and by applying Azuma-Hoeffding inequality, with probability at least $1 - \frac{\delta}{2}$:

$$\sum_{k \in \mathbb{I}} \sum_{h=1}^{H} \zeta_h^k \leq \sqrt{2KH^3 \log(2/\delta)} \leq 2H\sqrt{KH\iota}$$

By combining the upper of three terms in Equation (41), recall that $\beta = c \cdot dH \sqrt{\log(2dT/\delta)}, \beta' = c' \cdot dH \sqrt{\log(2dT/\delta)}$ we obtain:

$$\sum_{k \in \mathbb{I}} \hat{V}_1^k(s_1^k) - \check{V}_1^k(s_1^k) \leq H\beta\iota\sqrt{Kp} + H\beta'\iota\sqrt{Kp} + 2H\sqrt{KH\iota} = C' \cdot \sqrt{d^3 H^3 T \iota^2}$$

for some absolute constant $C'$.

Hence, the total regret is given by:

$$\sum_{k=1}^{K} \hat{V}_1^k(s_1^k) - \check{V}_1^k(s_1^k) \leq \frac{C'}{p} \cdot \sqrt{d^3 H^3 T \iota^2} = \tilde{O}(\sqrt{d^3 H^3 T \iota^2})$$

This concludes that the total pseudo regret of policy $\pi$ over $K$ episode is given by $\tilde{\mathcal{O}}(\sqrt{d^3 H^3 T \iota^2})$. And equivalently, we conclude that our algorithm obtains $\epsilon$-optimal policy with $\tilde{\mathcal{O}}(\frac{d^3 H^4}{\epsilon^2})$ samples with probability at least $1 - \delta$.

$\square$

## C. Auxiliary Lemmas

**Lemma C.1.** *(Jin et al., 2020) Let $\Lambda_t = \lambda \mathbf{I} + \sum_{i=1}^{t} \phi_i \phi_i^\top$ where $\phi_i \in \mathbb{R}^d$ and $\lambda > 0$. Then:*

$$\sum_{i=1}^{t} \phi_i^\top (\Lambda_t)^{-1} \phi_i \leq d$$

*Proof.* We have $\sum_{i=1}^{t} \phi_i^\top (\Lambda_t)^{-1} \phi_i = \sum_{i=1}^{t} \text{tr} \left( \phi_i^\top (\Lambda_t)^{-1} \phi_i \right) = \text{tr} \left( (\Lambda_t)^{-1} \sum_{i=1}^{t} \phi_i \phi_i^\top \right)$. Given the eigen-value decomposition $\sum_{i=1}^{t} \phi_i \phi_i^\top = \mathbf{U} \text{diag}(\lambda_1, \ldots, \lambda_d) \mathbf{U}^\top$, we have $\Lambda_t = \mathbf{U} \text{diag}(\lambda_1 + \lambda, \ldots, \lambda_d + \lambda) \mathbf{U}^\top$, and $\text{tr} \left( (\Lambda_t)^{-1} \sum_{i=1}^{t} \phi_i \phi_i^\top \right) = \sum_{j=1}^{d} \lambda_j / (\lambda_j + \lambda) \leq d$.content... $\square$

**Lemma C.2.** *(Abbasi-Yadkori et al., 2011) Let $\{\phi_t\}_{t \geq 0}$ be a bounded sequence in $\mathbb{R}^d$ satisfying $\sup_{t \geq 0} \|\phi_t\| \leq 1$. Let $\Lambda_0 \in \mathbb{R}^{d \times d}$ be a positive definite matrix. For any $t \geq 0$, we define $\Lambda_t = \Lambda_0 + \sum_{j=1}^{t} \phi_j \phi_j^\top$. Then, if the smallest eigenvalue of $\Lambda_0$ satisfies $\lambda_{\min}(\Lambda_0) \geq 1$, we have*

$$\log \left[ \frac{\det(\Lambda_t)}{\det(\Lambda_0)} \right] \leq \sum_{j=1}^{t} \phi_j^\top \Lambda_{j-1}^{-1} \phi_j \leq 2 \log \left[ \frac{\det(\Lambda_t)}{\det(\Lambda_0)} \right]$$

*Proof.* Since $\lambda_{\min}(\Lambda_0) \geq 1$ and $\|\phi_t\| \leq 1$ for all $j \geq 0$, we have

$$\phi_j^\top \Lambda_{j-1}^{-1} \phi_j \leq [\lambda_{\min}(\Lambda_0)]^{-1} \cdot \|\phi_j\|^2 \leq 1, \quad \forall j \geq 0.$$

Note that, for any $x \in [0, 1]$, it holds that $\log(1 + x) \le x \le 2\log(1 + x)$. Therefore, we have

$$\sum_{j=1}^{t} \log\left(1 + \phi_j^\top \Lambda_{j-1}^{-1} \phi_j\right) \le \sum_{j=1}^{t} \phi_j^\top \Lambda_{j-1}^{-1} \phi_j \le 2\sum_{j=1}^{t} \log\left(1 + \phi_j^\top \Lambda_{j-1}^{-1} \phi_j\right) \tag{48}$$

Moreover, for any $t \ge 0$, by the definition of $\Lambda_t$, we have

$$\det\left(\Lambda_t\right) = \det\left(\Lambda_{t-1} + \phi_t \phi_t^\top\right) = \det\left(\Lambda_{t-1}\right) \cdot \det\left(\mathbf{I} + \Lambda_{t-1}^{-1/2} \phi_t \phi_t^\top \Lambda_{t-1}^{-1/2}\right)$$

Since $\det\left(\mathbf{I} + \Lambda_{t-1}^{-1/2} \phi_t \phi_t^\top \Lambda_{t-1}^{-1/2}\right) = 1 + \phi_t^\top \Lambda_{t-1}^{-1} \phi_t$, the recursion gives:

$$\sum_{j=1}^{t} \log\left(1 + \phi_j^\top \Lambda_{j-1}^{-1} \phi_j\right) = \log \det\left(\Lambda_t\right) - \log \det\left(\Lambda_0\right) \tag{49}$$

Therefore, combining Equation (48) and Equation (49), we conclude the proof. $\square$

In our algorithm, full-exploratory trajectory occasionally occurs, and other trajectories also contributes our parameter $\Lambda_h^k$, in the following Lemma, we show that by adding more data, the bound remains effective.

**Lemma C.3.** *Let $\{\phi_t\}_{t \ge 0}$ be a bounded sequence in $\mathbb{R}^d$ satisfying $\sup_{t \ge 0} \|\phi_t\| \le 1$. And let $\{\psi_s\}_{s \ge 0}$ be another sequence of in $\mathbb{R}^d$ satisfying $\sup_{s \ge 0} \|\psi_s\| \le 1$. Let $\Lambda_0 \in \mathbb{R}^{d \times d}$ be a positive definite matrix. For any $t \ge 0$, $s \ge 0$, we define $\Lambda_t = \Lambda_0 + \sum_{j=1}^{t} \phi_j \phi_j^\top$, $\Lambda_{t,s} = \Lambda_0 + \sum_{j=1}^{t} \phi_j \phi_j^\top + \sum_{i=1}^{s} \psi_i \psi_i^\top$. Then, if the smallest eigenvalue of $\Lambda_0$ satisfies $\lambda_{\min}\left(\Lambda_0\right) \ge 1$, we have*

$$\sum_{j=1}^{t} \phi_j^\top \Lambda_{j-1,s_j}^{-1} \phi_j \le 2\log\left[\frac{\det\left(\Lambda_t\right)}{\det\left(\Lambda_0\right)}\right]$$

*where $\{s_j\}_{1 \le j \le t}$ is any non-decreasing sequence of number satisfying $s_j \in \mathbb{N}$.*

*Proof.* Consider any $t, s \in \mathbb{N}$, since $\Lambda_0$ is positive definite, and $\sum_{j=1}^{t} \phi_j \phi_j^\top$ and $\sum_{i=1}^{s} \psi_i \psi_i^\top$ are semi-positive-definite, we know that $\sigma(\Lambda_{t,s}) \ge \sigma(\Lambda_t)$ and $\sigma(\Lambda_t^{-1}) \ge \sigma(\Lambda_{t,s}^{-1})$ in a pointwise manner. This gives us, for any sequence $\{s_j\}_{1 \le j \le t}$, $s_j \in \mathbb{N}$,

$$\sum_{j=1}^{t} \phi_j^\top \Lambda_{j-1,s_j}^{-1} \phi_j \le \sum_{j=1}^{t} \phi_j^\top \Lambda_{j-1}^{-1} \phi_j \le 2\log\left[\frac{\det\left(\Lambda_t\right)}{\det\left(\Lambda_0\right)}\right]$$

This concludes the proof.

$\square$

**Lemma C.4.** *(Concentration of Self-Normalized Processes (Abbasi-Yadkori et al., 2011)). Let $\{\varepsilon_t\}_{t=1}^{\infty}$ be a real-valued stochastic process with corresponding filtration $\{\mathcal{F}_t\}_{t=0}^{\infty}$. Let $\varepsilon_t \mid \mathcal{F}_{t-1}$ be zero-mean and $\sigma$-subGaussian; i.e. $\mathbb{E}\left[\varepsilon_t \mid \mathcal{F}_{t-1}\right] = 0$, and*

$$\forall \lambda \in \mathbb{R}, \quad \mathbb{E}\left[e^{\lambda \varepsilon_t} \mid \mathcal{F}_{t-1}\right] \le e^{\lambda^2 \sigma^2 / 2}.$$

*Let $\{_t\}_{t=0}^{\infty}$ be an $\mathbb{R}^d$-valued stochastic process where $\phi_t \in \mathcal{F}_{t-1}$. Assume $\Lambda_0$ is a $d \times d$ positive definite matrix, and let $\Lambda_t = \Lambda_0 + \sum_{s=1}^{t} \phi_s \phi_s^\top$. Then for any $\delta > 0$, with probability at least $1 - \delta$, we have for all $t \ge 0$ :*

$$\left\| \sum_{s=1}^{t} \phi_s \varepsilon_s \right\|_{\Lambda_t^{-1}}^2 \leq 2\sigma^2 \log \left[ \frac{\det\left(\Lambda_t\right)^{1/2} \det\left(\Lambda_0\right)^{-1/2}}{\delta} \right]$$

**Lemma C.5.** *(Jin et al., 2020) Let $\{s_\tau\}_{\tau=1}^{\infty}$ be a stochastic process on state space $\mathcal{S}$ with corresponding filtration $\{\mathcal{F}_\tau\}_{\tau=0}^{\infty}$. Let $\{\phi_\tau\}_{\tau=0}^{\infty}$ be an $\mathbb{R}^d$-valued stochastic process where $\phi_\tau \in \mathcal{F}_{\tau-1}$, and $\|\phi_\tau\| \leq 1$. Let $\Lambda_k = \lambda I + \sum_{\tau=1}^{k} \phi_\tau \phi_\tau^\top$. Then for any $\delta > 0$, with probability at least $1 - \delta$, for all $k \geq 0$, and any $V \in \mathcal{V}$ so that $\sup_s |V(s)| \leq H$, we have:*

$$\left\| \sum_{\tau=1}^{k} \phi_\tau \left\{ V\left(s_\tau\right) - \mathbb{E}\left[V\left(s_\tau\right) \mid \mathcal{F}_{\tau-1}\right] \right\} \right\|_{\Lambda_k^{-1}}^2 \leq 4H^2 \left[ \frac{d}{2} \log\left( \frac{k+\lambda}{\lambda} \right) + \log \frac{\mathcal{N}_\varepsilon}{\delta} \right] + \frac{8k^2 \varepsilon^2}{\lambda},$$

*where $\mathcal{N}_\varepsilon$ is the $\varepsilon$-covering number of $\mathcal{V}$ with respect to the distance $\text{dist}(V, V') = \sup_s |V(s) - V'(s)|$.*

*Proof.* For any $V \in \mathcal{V}$, we know there exists a $\tilde{V}$ in the $\varepsilon$-covering such that

$$V = \widetilde{V} + \Delta_V \quad \text{and} \quad \sup_s |\Delta_V(s)| \leq \varepsilon$$

This gives following decomposition:

$$\left\| \sum_{\tau=1}^{k} \phi_\tau \left\{ V\left(s_\tau\right) - \mathbb{E}\left[V\left(s_\tau\right) \mid \mathcal{F}_{\tau-1}\right] \right\} \right\|_{\Lambda_k^{-1}}^2 \tag{50}$$

$$\leq 2 \left\| \sum_{\tau=1}^{k} \phi_\tau \left\{ \widetilde{V}\left(s_\tau\right) - \mathbb{E}\left[\widetilde{V}\left(s_\tau\right) \mid \mathcal{F}_{\tau-1}\right] \right\} \right\|_{\Lambda_k^{-1}}^2 + 2 \left\| \sum_{\tau=1}^{k} \phi_\tau \left\{ \Delta_V\left(s_\tau\right) - \mathbb{E}\left[\Delta_V\left(s_\tau\right) \mid \mathcal{F}_{\tau-1}\right] \right\} \right\|_{\Lambda_k^{-1}}^2, \tag{51}$$

where we can apply Theorem D.3 and a union bound to the first term. Also, it is not hard to bound the second term by $8k^2 \varepsilon^2 / \lambda$.

To compute the covering number of function class $\mathcal{V}$, we first require a basic result on the covering number of a Euclidean ball as follows. We refer readers to classical material, such as Lemma 5.2 in [44], for its proof. Lemma D.5 (Covering Number of Euclidean Ball). For any $\varepsilon > 0$, the $\varepsilon$-covering number of the Euclidean ball in $\mathbb{R}^d$ with radius $R > 0$ is upper bounded by $(1 + 2R/\varepsilon)^d$. □

**Lemma C.6.** *(Covering Number of Euclidean Ball). For any $\varepsilon > 0$, the $\varepsilon$-covering number of the Euclidean ball in $\mathbb{R}^d$ with radius $R > 0$ is upper bounded by $(1 + 2R/\varepsilon)^d$.*

Based on the lemmas above, we can bound the covering number of the optimistic value function and pessimistic value function class.

**Lemma C.7.** *(Covering number of optimistic function class (Jin et al., 2020)) Let $\mathcal{V}$ denote a class of functions mapping from $\mathcal{S}$ to $\mathbb{R}$ with following parametric form*

$$V(\cdot) = \min\left\{ \max_a \mathbf{w}^\top \phi(\cdot, a) + \beta \sqrt{\phi(\cdot, a)^\top \Lambda^{-1} \phi(\cdot, a)}, H \right\}$$

*where the parameters $(\mathbf{w}, \beta, \Lambda)$ satisfy $\|\mathbf{w}\| \leq L, \beta \in [0, B]$ and the minimum eigenvalue satisfies $\lambda_{\min}(\Lambda) \geq \lambda$. Assume $\|\phi(s, a)\| \leq 1$ for all $(s, a)$ pairs, and let $\mathcal{N}_\varepsilon$ be the $\varepsilon$-covering number of $\mathcal{V}$ with respect to the distance $\text{dist}(V, V') = \sup_s |V(s) - V'(s)|$. Then*

$$\log \mathcal{N}_\varepsilon \leq d \log(1 + 4L/\varepsilon) + d^2 \log\left[ 1 + 8d^{1/2} B^2 / \left(\lambda \varepsilon^2\right) \right]$$

*Proof.* Equivalently, we can reparametrize the function class $\mathcal{V}$ by let $\mathbf{A} = \beta^2 \Lambda^{-1}$, so we have

$$V(\cdot) = \min \left\{ \max_a \mathbf{w}^\top \phi(\cdot, a) + \sqrt{\phi(\cdot, a)^\top \mathbf{A} \phi(\cdot, a)}, H \right\} \tag{52}$$

for $\|\mathbf{w}\| \leq L$ and $\|\mathbf{A}\| \leq B^2 \lambda^{-1}$. For any two functions $V_1, V_2 \in \mathcal{V}$, let them take the form in Equation (52) with parameters $(\mathbf{w}_1, \mathbf{A}_1)$ and $(\mathbf{w}_2, \mathbf{A}_2)$, respectively. Then, since both $\min\{\cdot, H\}$ and $\max_a$ are contraction maps, we have

$$\text{dist}\,(V_1, V_2) \leq \sup_{s,a} \left\| \left[ \mathbf{w}_1^\top \phi(s, a) + \sqrt{\phi(s, a)^\top \mathbf{A}_2 \phi(s, a)} \right] - \left[ \mathbf{w}_2^\top \phi(s, a) + \sqrt{\phi(s, a)^\top \mathbf{A}_2 \phi(s, a)} \right] \right\| \tag{53}$$

$$\leq \sup_{\phi : \|\phi\| \leq 1} \left\| \left[ \mathbf{w}_1^\top \phi + \sqrt{\phi^\top \mathbf{A}_2 \phi} \right] - \left[ \mathbf{w}_2^\top \phi + \sqrt{\phi^\top \mathbf{A}_2 \phi} \right] \right\| \tag{54}$$

$$\leq \sup_{\phi : \|\phi\| \leq 1} \left| (\mathbf{w}_1 - \mathbf{w}_2)^\top \phi \right| + \sup_{\phi : \|\phi\| \leq 1} \sqrt{\left| \phi^\top (\mathbf{A}_1 - \mathbf{A}_2) \phi \right|} \tag{55}$$

$$= \|\mathbf{w}_1 - \mathbf{w}_2\| + \sqrt{\|\mathbf{A}_1 - \mathbf{A}_2\|} \leq \|\mathbf{w}_1 - \mathbf{w}_2\| + \sqrt{\|\mathbf{A}_1 - \mathbf{A}_2\|_F} \tag{56}$$

where the second last inequality follows from the fact that $|\sqrt{x} - \sqrt{y}| \leq \sqrt{|x - y|}$ holds for any $x, y \geq 0$. For matrices, $\| \cdot \|$ and $\| \cdot \|_F$ denote the matrix operator norm and Frobenius norm respectively.

Let $\mathcal{C}_{\mathbf{w}}$ be an $\varepsilon/2$-cover of $\left\{ \mathbf{w} \in \mathbb{R}^d \mid \|\mathbf{w}\| \leq L \right\}$ with respect to the 2-norm, and $\mathcal{C}_{\mathbf{A}}$ be an $\varepsilon^2/4$-cover of $\left\{ \mathbf{A} \in \mathbb{R}^{d \times d} \mid \|\mathbf{A}\|_F \leq d^{1/2} B^2 \lambda^{-1} \right\}$ with respect to the Frobenius norm. By Lemma C.6, we know:

$$|\mathcal{C}_{\mathbf{w}}| \leq (1 + 4L/\varepsilon)^d, \quad |\mathcal{C}_{\mathbf{A}}| \leq \left[ 1 + 8d^{1/2} B^2 / \left( \lambda \varepsilon^2 \right) \right]^{d^2}$$

By Equation (53), for any $V_1 \in \mathcal{V}$, there exists $\mathbf{w}_2 \in \mathcal{C}_{\mathbf{w}}$ and $\mathbf{A}_2 \in \mathcal{C}_{\mathbf{A}}$ such that $V_2$ parametrized by $(\mathbf{w}_2, \mathbf{A}_2)$ satisfies $\text{dist}\,(V_1, V_2) \leq \varepsilon$. Hence, it holds that $\mathcal{N}_\varepsilon \leq |\mathcal{C}_{\mathbf{w}}| \cdot |\mathcal{C}_{\mathbf{A}}|$, which gives:

$$\log \mathcal{N}_\varepsilon \leq \log |\mathcal{C}_{\mathbf{w}}| + \log |\mathcal{C}_{\mathbf{A}}| \leq d \log(1 + 4L/\varepsilon) + d^2 \log \left[ 1 + 8d^{1/2} B^2 / \left( \lambda \varepsilon^2 \right) \right]$$

This concludes the proof. $\qquad \square$

And we can obtain the same covering number bound on our pessimistic value function class due to the symmetry.

**Lemma C.8.** *(Covering number of pessimistic function class) Let $\mathcal{V}$ denote a class of functions mapping from $\mathcal{S}$ to $\mathbb{R}$ with following parametric form*

$$V(\cdot) = \text{clip} \left( \max_a \mathbf{w}^\top \phi(\cdot, a) - \beta \sqrt{\phi(\cdot, a)^\top \Lambda^{-1} \phi(\cdot, a)}, 0, H \right)$$

*where the parameters $(\mathbf{w}, \beta, \Lambda)$ satisfy $\|\mathbf{w}\| \leq L, \beta \in [0, B]$ and the minimum eigenvalue satisfies $\lambda_{\min}(\Lambda) \geq \lambda$. Assume $\|\phi(s, a)\| \leq 1$ for all $(s, a)$ pairs, and let $\mathcal{N}_\varepsilon$ be the $\varepsilon$-covering number of $\mathcal{V}$ with respect to the distance $\text{dist}\,(V, V') = \sup_s |V(s) - V'(s)|$. Then*

$$\log \mathcal{N}_\varepsilon \leq d \log(1 + 4L/\varepsilon) + d^2 \log \left[ 1 + 8d^{1/2} B^2 / \left( \lambda \varepsilon^2 \right) \right]$$

*Proof.* Similar to the proof strategy used in Lemma C.7, we reparametrize the function the function class $\mathcal{V}$ by letting $\mathbf{A} = \beta^2 \Lambda^{-1}$, which gives us,

$$V(\cdot) = \text{clip}\left(\max_a \mathbf{w}^\top \phi(\cdot,a) - \sqrt{\phi(\cdot,a)^\top \mathbf{A}\phi(\cdot,a)}, 0, H\right) \tag{57}$$

for $\|\mathbf{w}\| \leq L$ and $\|\mathbf{A}\| \leq B^2\lambda^{-1}$. For any two functions $V_1, V_2 \in \mathcal{V}$, let them take the form in Equation (57) with parameters $(\mathbf{w}_1, \mathbf{A}_1)$ and $(\mathbf{w}_2, \mathbf{A}_2)$, respectively. Then, since both $\text{clip}(\cdot, 0, H)$ and $\max_a$ are contraction maps, we have

$$\text{dist}\,(V_1, V_2) \leq \sup_{s,a} \left|\left[\mathbf{w}_1^\top \phi(s,a) - \sqrt{\phi(s,a)^\top \mathbf{A}_1 \phi(s,a)}\right] - \left[\mathbf{w}_2^\top \phi(s,a) - \sqrt{\phi(s,a)^\top \mathbf{A}_2 \phi(s,a)}\right]\right| \tag{58}$$

$$\leq \sup_{\phi:\|\phi\|\leq 1} \left|\left[\mathbf{w}_1^\top \phi - \sqrt{\phi^\top \mathbf{A}_1 \phi}\right] - \left[\mathbf{w}_2^\top \phi - \sqrt{\phi^\top \mathbf{A}_2 \phi}\right]\right| \tag{59}$$

$$\leq \sup_{\phi:\|\phi\|\leq 1} \left|(\mathbf{w}_1 - \mathbf{w}_2)^\top \phi\right| + \sup_{\phi:\|\phi\|\leq 1} \sqrt{\left|\phi^\top (\mathbf{A}_2 - \mathbf{A}_1)\phi\right|} \tag{60}$$

$$= \|\mathbf{w}_1 - \mathbf{w}_2\| + \sqrt{\|\mathbf{A}_2 - \mathbf{A}_1\|} \leq \|\mathbf{w}_1 - \mathbf{w}_2\| + \sqrt{\|\mathbf{A}_2 - \mathbf{A}_1\|_F} \tag{61}$$

$$= \|\mathbf{w}_1 - \mathbf{w}_2\| + \sqrt{\|\mathbf{A}_1 - \mathbf{A}_2\|_F} \tag{62}$$

Equation (58) shows that the distance of two elements in pessimistic value function class shares a same upper bound with the optimistic value function class. By Lemma C.7, we conclude the proof. $\square$

