# OpenReview forum: "Hyper: Hyperparameter Robust Efficient Exploration in Reinforcement Learning"
_ICML.cc/2025/Conference — ICML 2025 poster_

### Official Review · Reviewer_QGsd · 2025-03-03

**Overall Recommendation:** 3

**Summary:**

This submission proposes a novel method, referred to as Hyper, to address the challenging issue of hyperparameter tuning in curiosity-based exploration methods. It introduces a repositioning and exploration mechanism that controls the horizon of exploitation before conducting exploration. The length of the exploitation horizon is sampled from a bounded Geometric distribution. The authors provide both theoretical and empirical evidence to demonstrate the effectiveness of the proposed Hyper method.

**Claims And Evidence:**

The claims made in the submission are supported by clear and convincing evidence.

**Essential References Not Discussed:**

This submission includes sufficient related references.

**Experimental Designs Or Analyses:**

The experimental design and analysis exhibit soundness and validity.

**Methods And Evaluation Criteria:**

The proposed methods and the corresponding evaluation criteria are appropriate for addressing the problem.

**Other Comments Or Suggestions:**

1. On line 175 of the right column, it is mentioned "We defer the formal proof to the appendix." The specific appendix should be cited for clarity.

**Other Strengths And Weaknesses:**

**Strengths**
1. Addressing the exploration-exploitation dilemma in reinforcement learning is valuable.
2. The proposed Hyper method is well-introduced and clearly described.
3. The authors provide theoretical analysis and empirical studies supporting the effectiveness of the proposed methods.

**Weaknesses**
See the following comments and questions.

**Questions For Authors:**

1. I do not see the efficiency of Hyper according to the upper bound involving the $d^3H^4$. Could you clarify?
2. What is the $\epsilon$ in Theorem 4.2?
3. I see that the proposed Hyper method is an improvement based on curiosity-based methods. How does it compare to more recent exploration algorithms?
4. Is the parameter $\gamma$, which controls the probability of the repositioning phase, fixed for all tasks?

**Relation To Broader Scientific Literature:**

This work primarily focuses on the field of reinforcement learning.

**Theoretical Claims:**

The proofs for the theoretical claims in this submission are correct.

---

> ### Author Rebuttal · Authors · 2025-04-01
>
> Thank you for your positive assessment and insightful questions. We appreciate your recognition of our work's value and address your questions below.
>
> ***Regarding Hyper’s efficiency***
>
> Our claim that "Hyper is efficient" operates on two complementary levels:
>
> 1. **Theoretical guarantees**: Hyper is provably guaranteed to sufficiently explore the environment and converge to an optimal policy, as demonstrated by our theoretical analysis providing a worst-case sample complexity bound.
> 2. **Empirical efficiency**: Hyper demonstrates significantly greater empirical efficiency and robustness to the curiosity coefficient $\beta$ compared to baseline methods, as conclusively shown in our experimental evaluation in Section 6.
>
> While our theoretical sample complexity appears similar to existing methods in the worst case, our empirical results consistently demonstrate superior practical efficiency. This pattern of theoretical bounds appearing similar while practical performance differs substantially, this is common in reinforcement learning research, where worst-case bounds often don't fully capture the advantages of sophisticated exploration strategies in real environments.
>
> ***Regarding $\epsilon$ in Theorem 4.2***
>
> It represents the optimality gap: the maximum distance between the value of the current policy and the optimal policy. Specifically, a policy $\pi$ is $\epsilon$-optimal if it satisfies: $V^*(s) - V^{\pi}(s) < \epsilon, \forall s \in \mathcal{S}$. This is standard notation in RL theory.
>
> ***Regarding comparison to recent methods***
>
> Curiosity-based exploration remains the dominant paradigm in the field due to its empirical effectiveness, particularly in sparse-reward environments. Hyper is designed as a general framework compatible with any off-policy RL algorithm and any curiosity method. This extensible design allows Hyper to leverage advances in both RL algorithms and curiosity methods, ensuring its continued relevance as the field progresses.
>
> In our main experiments (Section 6), we use TD3 as the RL algorithm and Disagreement as the curiosity method for all methods (including baselines) for fair comparison. Additionally, in Appendix A.5, we compare with LESSON, an advanced recent method. For this comparison, we follow LESSON's original implementation using DQN as the RL algorithm and RND as the curiosity method. Hyper consistently outperforms LESSON on the MiniGrid environments used in their original paper, demonstrating its state-of-the-art performance.
>
> ***Regarding truncation probability parameter $\gamma$ and $p$***
>
> The discount factor $\gamma=0.99$ is fixed across all environments, as is standard in RL research. For the truncation probability $p$, we use a unified decay schedule from 0.01 to 0.001 across all environments, as described in Section 5.2 and Appendix A.1.
>
> This consistent parameterization across diverse environments highlights Hyper's robustness—it maintains strong performance without environment-specific tuning, unlike traditional curiosity-driven methods that require extensive hyperparameter adjustment for each new environment.

---

> > ### Comment · Reviewer_QGsd · 2025-04-04
> >
> > Thank you for your thoughtful response. I have no further concerns and will maintain my current score.

---

> > > ### Author Response · Authors · 2025-04-07
> > >
> > > Thank you for your consistent support of our work.
> > >
> > > We're pleased that our rebuttal has comprehensively addressed all of your concerns and questions. Given that all reviewers now acknowledge the value of our contribution and we've successfully resolved all initial concerns, we would kindly request you consider raising your score to better reflect the significant contribution our paper makes to the RL community. We appreciate your thoughtful evaluation throughout this process.

---

### Official Review · Reviewer_Sbxf · 2025-03-11

**Overall Recommendation:** 3

**Summary:**

This paper proposes a “repositioning_length” based method to alternate between exploration and explosion. The key idea is to choose the bounded geometric distribution with probability p to determine the repositioning_length to make the process more sample efficient.

**Claims And Evidence:**

Yes the motivation is clear though the presentation of method is unclear.

**Essential References Not Discussed:**

No to my knowledge.

**Experimental Designs Or Analyses:**

Yes.

**Methods And Evaluation Criteria:**

Yes it uses toy tasks and some basic robot related tasks.

**Other Comments Or Suggestions:**

- In Fig. 1, you should illustrate what β represents.
- The first equation in Section 3 is unclear, as there is no indication that a discount is applied to the intrinsic reward.
Use vector graphics for the figure plots.
- The title "Hyperparameter Robust Exploration in Reinforcement Learning" is quite broad. To make it more explicit, consider including terms like "Repositioning & Exploration" for greater clarity.
- Lines 16 and 17 in Algorithm 1 are unclear. It is not evident whether the method uses only one type of reward or if both types of rewards are used in each phase.

**Other Strengths And Weaknesses:**

# Strengths:
- The presentation regarding motivation is generally clear, and the method is simple.
- Results show improvement in most empirical evaluations.
# Weaknesses:
- The writing quality of the paper is moderate. The core design choice revolves around the repositioning-and-exploration mechanism, which requires a balanced approach. The paper needs substantial revision to improve clarity and readability.
- Section 5.3 is the core of the paper. However, it is wordy and lacks informativeness, making it difficult to follow the method clearly. - Currently, Algorithm 1 provides the most concrete description of the method. The authors should refine Section 5.3 to offer a clearer explanation.
- In Algorithm 1, I cannot find β. I am wondering whether the proposed method explicitly incorporates β in its design loop. From Figures 1 and 6, it appears that β is part of the algorithm loop.
- The comparison with LESSON (another method for switching between exploration and exploitation) is questionable, as it is based on a single comparison in specific MiniGrid tasks.

**Questions For Authors:**

No.

**Relation To Broader Scientific Literature:**

It might inspire the understanding on the exploration.

**Theoretical Claims:**

No.

---

> ### Author Rebuttal · Authors · 2025-04-01
>
> We appreciate your feedback, though we must respectfully note that **your summary appears to significantly mischaracterize our paper's contributions and scope**. Your summary focuses narrowly on a single implementation detail (the bounded geometric distribution) without acknowledging our paper's core contribution: addressing the fundamental hyperparameter sensitivity problem in curiosity-driven exploration. While the other three reviewers correctly identified this primary contribution along with our theoretical analysis and empirical validation, your assessment seems to overlook these central aspects. While our algorithm is elegantly simple, its design required substantial analysis to identify why existing methods fail in certain cases and how our approach resolves these limitations. This fundamental misunderstanding appears to have colored your assessment, as evidenced by your subsequent questions. Nevertheless, we address your specific concerns below.
>
>
> ### Difficulty of Understanding the Meaning of $\beta$
>
> > In Algorithm 1, I cannot find $\beta$. I am wondering whether the proposed method explicitly incorporates $\beta$ in its design loop. From Figures 1 and 6, it appears that $\beta$ is part of the algorithm loop.
>
> > In Fig. 1, you should illustrate what $\beta$ represents.
>
> $\beta$ is the coefficient controlling the scale of curiosity reward, as explained in detail in Section 1 (Introduction). This is standard notation in the field. For clarity, we will explicitly include $\beta$ in Algorithm 1 in the camera-ready version, though it is implicitly present in the "with intrinsic reward" training step (line 16).
>
> The core contribution of our paper is precisely that Hyper significantly reduces sensitivity to $\beta$, as conclusively demonstrated in Figure 6. This addresses a central challenge in curiosity-driven exploration methods that has limited their practical applicability.
>
>
> ### Difficulty of Understanding Section 5
>
> > The writing quality of the paper is moderate. The core design choice revolves around the repositioning-and-exploration mechanism, which requires a balanced approach. The paper needs substantial revision to improve clarity and readability.
>
> > Section 5.3 is the core of the paper. However, it is wordy and lacks informativeness, making it difficult to follow the method clearly. - Currently, Algorithm 1 provides the most concrete description of the method. The authors should refine Section 5.3 to offer a clearer explanation.
>
> We will refine Section 5.3 for clarity while maintaining its informative content. It's worth noting that other reviewers (L37S, 3vGR, and QGsd) did not express difficulty understanding our method or Section 5, suggesting the explanation is generally effective. We will strengthen this section by more explicitly connecting the theoretical insights to the practical implementation.
>
>
>
> > The comparison with LESSON is questionable, as it is based on a single comparison in specific MiniGrid tasks.
>
> Our comparison with LESSON follows standard scientific practice by evaluating on the environments used in the original LESSON paper. Fetch8x8, UnlockPickup, and LavaCrossingS9N1 are specifically tasks where LESSON demonstrated its strong performance compared to previous option-based and curiosity-driven methods. Hyper's superior performance on these same tasks provides strong evidence of its effectiveness. This approach to comparison is fair, rigorous, and follows established standards in the field.
>
> > The first equation in Section 3 is unclear, as there is no indication that a discount is applied to the intrinsic reward. Use vector graphics for the figure plots.
>
> The discount is indeed applied to the intrinsic reward $b(s, a, s')$, as indicated by its placement within the brackets. This follows standard notation in the field. We will use vector graphics for all figures in the camera-ready version.
>
> > The title is quite broad. To make it more explicit, consider including terms like "Repositioning & Exploration" for greater clarity.
>
> The current title accurately reflects our paper's primary contribution: achieving hyperparameter robustness in RL exploration. "Repositioning & Exploration" is the mechanism we developed to achieve this goal, not the goal itself. The title appropriately emphasizes our main contribution rather than the specific technique used.
>
> > Lines 16 and 17 in Algorithm 1 are unclear. It is not evident whether the method uses only one type of reward or if both types of rewards are used in each phase.
>
> We will clarify this in the camera-ready version. "With intrinsic reward" means training uses both task reward and intrinsic reward together, while "Without intrinsic reward" means using task reward only.
>
>
> **We respectfully request that you reconsider your assessment in light of our clarifications above, as your review appears to have overlooked our paper's primary contribution that was correctly identified by all other reviewers.**

---

> > ### Comment · Reviewer_Sbxf · 2025-04-07
> >
> > Thank you to the authors for their patient explanation and kind response. The rebuttal addresses my concerns, and I am inclined to view this draft more favorably and will raise my score.

---

> > > ### Author Response · Authors · 2025-04-07
> > >
> > > We appreciate your decision to raise your score.
> > >
> > > We're pleased that our rebuttal has comprehensively addressed all of your concerns and questions. Given the strong theoretical guarantees and empirical results we've presented, and our commitment to incorporate all feedback in the camera-ready version, we respectfully hope you to further consider a stronger score.

---

### Official Review · Reviewer_3vGR · 2025-03-16

**Overall Recommendation:** 3

**Summary:**

The paper addresses hyperparameter sensitivity in curiosity-driven RL exploration, proposing **Hyper**, a two-phase algorithm that decouples exploration (curiosity-driven) and exploitation (repositioning-guided). Theoretical guarantees under linear MDP assumptions and empirical validation across navigation/locomotion tasks demonstrate Hyper’s robustness to the intrinsic reward coefficient β.

**Claims And Evidence:**

- **Claim 1 (β Sensitivity):** Supported by the navigation warm-up example (Fig. 2) and Table 1. **Limitation:** Baseline comparisons use β=1.0 for all methods, potentially disadvantaging baselines requiring tuned β.
- **Claim 2 (Hyper’s Robustness):** Empirical results (Fig. 6) validate reduced β sensitivity. **Gap:** Robustness to **p** (truncation probability) is not rigorously tested.
- **Claim 3 (Theoretical Efficiency):** Theorem 4.2 under linear MDPs is sound but lacks direct connection to practical Algorithm 1.

**Essential References Not Discussed:**

The paper covers most relevant literature, but could benefit from discussing:

1. **Meta-RL approaches to exploration**: Recent work on meta-learning exploration strategies (e.g., Stadie et al., "Some Considerations on Learning to Explore via Meta-Reinforcement Learning," 2018) could provide context for automatically adapting exploration strategies.

2. **Intrinsic motivation in hierarchical RL**: The paper could discuss connections to hierarchical RL methods that use intrinsic motivation (e.g., Kulkarni et al., "Hierarchical Deep Reinforcement Learning: Integrating Temporal Abstraction and Intrinsic Motivation," 2016).

3. **Exploration in partially observable environments**: Since many real-world tasks involve partial observability, discussing how Hyper might perform in such settings would be valuable.

**Experimental Designs Or Analyses:**

- **Statistical Significance:** Standard deviations reported but no formal tests (e.g., t-tests).
- **Metric Consistency:** Success rate (navigation) vs. cumulative reward (locomotion) is appropriate but obscures sample efficiency comparisons.

**Methods And Evaluation Criteria:**

- **Strengths:** Warm-up example effectively isolates β sensitivity; environment diversity (PointMaze, MuJoCo) tests multiple dimensions.
- **Weaknesses:**
  - **Baseline Fairness:** Using β=1.0 for all methods may skew comparisons (e.g., Curiosity-Driven methods often require smaller β).
  - **Ablation Studies:** Missing analysis of repositioning phase’s contribution vs. truncation mechanism.

**Other Comments Or Suggestions:**

1. The paper would benefit from a more detailed discussion of the limitations of Hyper, particularly cases where it might not perform well.
2. A visualization of the agent's behavior during the repositioning and exploration phases would help readers understand the algorithm's dynamics.
3. The connection between the theoretical algorithm (Algorithm 2) and the practical implementation (Algorithm 1) could be more clearly explained.
4. Minor typos:
   - In equation (1), "Qπ = Eπ[PHh=1 γh−1rh(sh, ah)]" should likely be "Qπ = Eπ[∑Hh=1 γh−1rh(sh, ah)]"
   - Several instances of missing or incorrect mathematical notation in the PDF rendering.

**Other Strengths And Weaknesses:**

**Strengths:**
1. The paper addresses a practical and significant problem in RL that limits the applicability of powerful exploration methods.
2. The proposed solution is elegant, combining theoretical guarantees with practical implementation.
3. The empirical results are comprehensive and convincing.
4. The paper is well-written and the ideas are clearly presented.

**Weaknesses:**
1. The practical implementation of Hyper still requires setting the truncation probability p, which introduces another hyperparameter. While the authors provide a reasonable default and decay schedule, this somewhat undermines the claim of hyperparameter robustness.
2. The analysis of why Hyper works is somewhat limited - more insight into the interaction between the repositioning phase and exploration would strengthen the paper.
3. The environments, while varied, are still relatively standard RL benchmarks. Testing on more diverse or challenging environments would strengthen the claims.

**Questions For Authors:**

1. **Truncation Probability Sensitivity:** How does Hyper’s performance vary with **p**? Does the decay schedule generalize across environments?
2. **Baseline Tuning:** Why was β=1.0 chosen for baselines? Were baselines tested with their optimal β ranges?
3. **Theory-Practice Gap:** How does Algorithm 1’s neural network implementation relate to Algorithm 2’s linear assumptions?

**Relation To Broader Scientific Literature:**

- **Strengths:** Builds on curiosity-driven (Pathak et al.) and decoupled RL (Schäfer et al.) literature.
- **Gaps:**
  - Fails to cite meta-RL exploration strategies (e.g., Stadie et al. 2018).
  - Omits hierarchical RL with intrinsic motivation (Kulkarni et al. 2016).

## Relation To Broader Scientific Literature

The paper is well-positioned within the broader literature on exploration in RL:

1. It builds upon the established curiosity-driven exploration methods (Bellemare et al., 2016; Pathak et al., 2017, 2019; Burda et al., 2018).

2. It addresses a practical limitation of these methods (hyperparameter sensitivity) that has been noted but not thoroughly addressed in previous work.

3. The decoupling of exploration and exploitation relates to work by Schäfer et al. (2021) and Whitney et al. (2021), but with the novel addition of the repositioning mechanism.

4. The theoretical analysis extends the line of work on provably efficient RL with function approximation (Jin et al., 2018, 2020; Yang & Wang, 2020).

**Theoretical Claims:**

- Theorem 4.2 (sample efficiency) is valid under linear MDPs but assumes Algorithm 2 (theoretical) aligns with Algorithm 1 (practical). **Gap:** No discussion of how neural networks in practice affect theoretical guarantees.

---

> ### Author Rebuttal · Authors · 2025-04-01
>
> Thank you for your detailed feedback, and we appreciate your positive comments. We will address your concerns below.
>
> ***Regarding relevant literatures***
>
> Thank you for the valuable suggestion. We will incorporate discussions on the references you suggested in the camera-ready version.
>
> ***Regarding partially observable environments***
>
> We would like to refer you to Figure 9 in our paper, which shows the experimental results on MiniGrid environments, where the agent observes limited field-of-view (partially observable) observations. Hyper outperforms LESSON by Kim et al. (2023) in MiniGrid (Figure 9), which itself was already shown superior to standard curiosity approaches. This suggests Hyper's mechanisms transfer effectively to partially observable settings.
>
> ***Regarding baseline fairness***
>
> We used $\beta=1.0$ following the official Disagreement implementation by Pathak et al. (2019) to ensure fair comparison with the baselines.
>
> ***Regarding experiment analysis and metric consistency***
>
> We believe that our experimental results can demonstrate Hyper's superior sample efficiency. The performance curves in Figures 5,8,9 show that Hyper consistently achieves faster learning across all environments compared to baseline methods. This demonstrates how quickly an algorithm reaches a given performance level, which is precisely the definition of sample efficiency in reinforcement learning.
>
> ***Regarding $p$-robustness***
>
> Hyper is much less sensitive to hyperparameters than existing methods as shown in Figure 6. Unlike $\beta$, the truncation probability $p$ requires no environment specific tuning. We use the same $p$ schedule across all experiments with consistently strong performance. Our truncated geometric distribution design ensures phase lengths adapt appropriately to different environment horizons. At extreme values, Hyper smoothly transitions between pure exploitation ($p=0$) and full exploration/Decouple ($p=1$).
>
> ***Regarding interaction between the repositioning and exploration phases***
>
> Section 5 comprehensively explains Hyper's key mechanisms:
> 1. The repositioning phase strategically places the agent in promising regions, preventing over-exploration demonstrated in our warm-up example
> 2. The truncated geometric distribution ensures sufficient exploration while focusing resources on promising areas
> 3. By limiting exploration to regions informed by the exploitation policy, Hyper collects data that aligns with the exploitation policy's distribution, preventing distribution shift problems (Figure 3) and enabling more efficient learning.
>
> ***Regarding the choices of environments***
>
> Our experiments demonstrate Hyper's performance across different challenges: diverse reward structures (dense-reward locomotion, sparse-reward locomotion, sparse-reward navigation), variable horizons (200-1000), and different state/action complexities. Hyper consistently excels across all settings, which provides compelling evidence for Hyper's generality and effectiveness.
>
> ***Regarding the limitations & future works***
>
> We appreciate this suggestion and will expand the limitations discussion in our revised paper. The primary limitations include:
>
> 1. **Computational cost**: Like Decouple, Hyper requires maintaining and training an additional exploitation policy, increasing computational overhead compared to traditional curiosity-driven methods.
> 2. **Potential for flexibility improvements**: While Hyper's exploration paradigm of alternating between repositioning and exploration phases proves highly effective, it can be further improved by dynamically switching between phases based on environment feedback.
>
> We will incorporate this into discussion in the camera-ready version.
>
> ***Regarding the visualization of agent behavior***
>
> Thank you for this valuable and insightful suggestion. We agree this visualization will strengthen the paper and have implemented it as suggested. The state-visitation plots can be found in this anonymous link: https://imgur.com/a/agent-behavior-8wQXtay.
>
> The visualizations clearly illustrate Hyper's advantage over baseline methods and provide further evidence of Hyper's exploration-exploitation balance.
>
> ***Regarding the connection between theory and practice***
>
> Hyper represents a general RL exploration paradigm that can be integrated with various off-policy algorithms and curiosity methods. Algo2 is a realization of Algo1 with linear function approximation and UCB intrinsic reward. Under necessary assumptions, our theoretical analysis for Algo2 demonstrates convergence guarantees under function approximation, while Algo 1 shows how the framework can be implemented with modern deep RL methods.
>
> ## Reference
>
> **Kim et al. (2023) LESSON: Learning to Integrate Exploration Strategies for Reinforcement Learning via an Option Framework. In ICML**
>
> **Pathak et al. (2019) Self-Supervised Exploration via Disagreement. In ICML**

---

> > ### Comment · Reviewer_3vGR · 2025-04-02
> >
> > Thank you for your detailed rebuttal addressing my concerns. After considering your responses, I have updated my assessment of your paper.
> >
> > ## Regarding Partially Observable Environments
> >
> > I appreciate the clarification about MiniGrid experiments in Figure 9. This indeed demonstrates Hyper's effectiveness in partially observable settings, which strengthens your claims about robustness across different environment types. The comparison against LESSON, which was already shown to outperform standard curiosity approaches, provides compelling evidence for Hyper's capabilities in this domain.
> >
> > ## Regarding Baseline Fairness
> >
> > Your explanation that β=1.0 follows the official Disagreement implementation by Pathak et al. (2019) addresses my concern about potential unfairness in the baseline comparisons. This adherence to established implementations strengthens the validity of your comparative results.
> >
> > ## Regarding Experiment Analysis and Metric Consistency
> >
> > The learning curves in Figures 5, 8, and 9 do indeed demonstrate Hyper's superior sample efficiency across environments. I agree that these results effectively show how quickly Hyper achieves given performance levels compared to baselines, which is a standard measure of sample efficiency in RL.
> >
> > ## Regarding p-robustness
> >
> > Your explanation about the truncation probability p is convincing. The fact that you used the same p schedule across all experiments with consistently strong performance is significant evidence of Hyper's robustness. The design of the truncated geometric distribution to adapt phase lengths to different environment horizons is a particularly elegant solution that addresses my concerns about introducing another hyperparameter.
> >
> > ## Regarding Interaction Between Phases
> >
> > Section 5 does provide a comprehensive explanation of Hyper's mechanisms. The visualization you've added (linked in the rebuttal) further clarifies how the repositioning phase strategically places the agent and prevents over-exploration. This visualization effectively demonstrates Hyper's advantage over baseline methods and helps explain why the approach works so well.
> >
> > ## Regarding Theory and Practice Connection
> >
> > Your explanation of the relationship between Algorithms 1 and 2 clarifies how the theoretical guarantees for the linear function approximation case relate to the practical implementation with deep RL methods. This addresses my concern about the gap between theory and practice in your approach.
> >
> > ## Regarding Limitations and Future Work
> >
> > I appreciate your commitment to expand the limitations discussion in the revised paper. The points you've identified about computational cost and potential flexibility improvements are important considerations for readers to understand the trade-offs involved in adopting your approach.
> >
> > ## **Regarding Related Work**
> >
> > I strongly suggest revisiting the discussion on Bayesian RL approaches to exploration. The current characterization of Bayesian RL is incomplete and fails to acknowledge more recent advances in this area. Please include discussion on:
> > ---
> >
> > 1. Osband, Ian, John Aslanides, and Albin Cassirer. "Randomized prior functions for deep reinforcement learning." Advances in neural information processing systems 31 (2018).
> >
> > 2. Osband, Ian, et al. "Deep exploration via randomized value functions." Journal of Machine Learning Research 20.124 (2019): 1-62.
> >
> > 3. Li, Yingru, et al. "Q-Star Meets Scalable Posterior Sampling: Bridging Theory and Practice via HyperAgent." International Conference on Machine Learning. PMLR, 2024.
> >
> > 4. Li, Yingru et al. “Scalable Thompson Sampling via Ensemble++ Agent.” (2024).
> >
> > These approaches have shown strong exploration capabilities in challenging environments, and a more thorough discussion would provide readers with a more accurate understanding of the current state of Bayesian exploration methods in RL.
> >
> > ## Updated Assessment
> >
> > Based on your responses and the additional materials provided, I now have a more positive view of your paper. The comprehensive experiments across diverse environments (including partially observable ones), the theoretical guarantees, and the clear explanation of Hyper's mechanisms make a compelling case for its effectiveness and robustness.
> >
> > The paper addresses an important practical problem in RL (hyperparameter sensitivity) with an elegant solution that is both theoretically grounded and empirically validated. The additional visualizations and clarifications you've provided further strengthen the paper's contributions.
> >
> > I recommend acceptance of this paper, as it makes a valuable contribution to the field of reinforcement learning by addressing a significant limitation of curiosity-driven exploration methods.

---

> > > ### Author Response · Authors · 2025-04-07
> > >
> > > Thank you for your positive feedback and your recommendation of acceptance!
> > >
> > > We have comprehensively addressed all your concerns and questions in our rebuttal, demonstrating the robustness and effectiveness of our approach. As suggested, we will enhance our paper with additional discussions on meta-exploration and Bayesian RL approaches, incorporating the valuable references you provided. Given our thorough responses and commitment to these improvements in the camera-ready version, we respectfully hope you consider further raising your score to better reflect the significant contribution our paper makes to the field.

---

### Official Review · Reviewer_L37S · 2025-03-16

**Overall Recommendation:** 3

**Summary:**

This paper has proposed a new method, referred to as "hyper-parameter robust exploration (Hyper)", which aims to mitigate the "extensive hyper-parameter tuning" problem in existing curiosity-based exploration methods. The proposed method Hyper is summarized in Algorithm 1. This paper also analyzes Hyper under the linear MDP setting (Theorem 4.2), and preliminary experiment results are demonstrated in Section 6. In particular, Section 6.2 demonstrates Hyper's robustness to $\beta$.

## update after rebuttal

I have read the rebuttal and discussed with the authors.

**Claims And Evidence:**

Overall, the main claims of this paper are supported by both the theoretical analysis (Theorem 4.2) and experiment results in Section 6. Some comments:

- The theoretical analysis and result are limited to linear MDPs. This is an obvious limitation, however, it is mainly due to existing analysis techniques in the theoretical RL community. I do not see an easy way to extend the analysis beyond the linear MDP framework.

- The existing experiment results in Section 6 are solid, but I am wondering if they can be further strengthened. Specifically, my understanding is that the proposed Hyper method is a general method for all curiosity-based exploration approaches. However, in Section 6, only experiment results under a few algorithms have been demonstrated. I recommend the authors to add more experiment results under more algorithms to further strengthen the paper.

**Essential References Not Discussed:**

I have not found any essential references that have not been discussed.

**Experimental Designs Or Analyses:**

I have checked the experiment design. To the best of my knowledge, it is sound and valid.

**Methods And Evaluation Criteria:**

The proposed methods and evaluation criteria make sense for the considered problem.

**Other Comments Or Suggestions:**

- Typo: in Section 2, when defining the total reward, $\gamma^{h-1}$ is missing before $b_h$

- This paper considers a setting with both the finite time horizon $H$ and the discount factor $\gamma$, this seems to be a non-standard RL setting. Usually we either consider a finite-horizon setting with $\gamma=1$, or an infinite horizon setting with $\gamma<1$.

**Other Strengths And Weaknesses:**

- The flow of this paper can be further improved. In particular, Algorithm 1 is after the analysis section (Section 4), which makes it a little bit difficult to read Section 4.

- I think the key points of Section 3 are well known to experts in this field. Maybe the authors can shorten it a little bit, and use the space to add more experiment results.

**Questions For Authors:**

Please try to address the weaknesses and questions listed above.

**Relation To Broader Scientific Literature:**

My understanding is that this paper has done a good job of literature review and well positions itself among the relevant literature.

**Theoretical Claims:**

I think the theoretical claims in Theorem 4.2 can be further strengthened, specifically

- Please discuss the tightness of the upper bound developed in Theorem 4.2. Ideally, this paper should also develop a lower bound under the linear MDP setting and discuss the tightness.

- Rather than the number of steps, why not present the results of Theorem 4.2 using a regret bound?

- Please discuss how the results depend on the truncation probability $p$. Currently they are hidden in the $\tilde{O}$ notation.

---

> ### Author Rebuttal · Authors · 2025-04-01
>
> Thank you for recognizing the strengths of our work, particularly our careful experiment design and comprehensive literature review.
>
> Regarding your concerns about theoretical aspects, we want to clarify that the theoretical analysis serves as a convergence guarantee. Hyper's exploration framework deliberately diverges from traditional curiosity-driven exploration approaches by utilizing an exploitive policy for part of the buffer collection. The LinearMDP framework analysis demonstrates that even with this novel approach, we achieve robust exploration efficiency in worst-case scenarios. We address your specific points below.
>
> ***Regarding truncation probability $p$ dependency in sample complexity***
>
> The sample complexity result is explicitly $O\left(\frac{d^3 H^4}{\epsilon^2 p}\right)$, which clearly shows how $p$ affects theoretical performance. This parameter provides valuable flexibility in exploration-exploitation balance:
>
> - When $p = 0$: The algorithm operates exclusively in exploitation mode. This will bring the theoretical sample complexity to infinity.
> - When $p = 1$: The algorithm commits fully to exploratory data collection, effectively functioning as Decouple algorithm. As Decouple algorithm exclusively uses exploration policy to collect the data, the sample complexity reduces to $O\left(\frac{d^3 H^4}{\epsilon^2 }\right)$, which exactly matches the bound in Jin et al. (2020).
>
> ***Regarding the tightness of the bound***
>
> Our upper bound matches bound in Jin et al. (2020). Regarding tightness, recent work by He et al. (2023) established a minimax-optimal bound of $O(d\sqrt{TH^3})$ for the linear setting, which aligns with the lower bound presented by Zhou et al. (2021). Our follow-up work is already exploring the integration of He et al.'s techniques to derive tighter bounds for Hyper, which would further strengthen our theoretical guarantees.
>
> ***Regarding the regret bound***
>
> Our sample complexity result derives directly from the regret bound $\tilde{O}(\sqrt{d^3 H^4 T \iota^2})$, or $O(\frac{\sqrt{d^3 H^4 T \iota^2}}{p})$, currently presented in the appendix. We will incorporate this regret bound into the main theorem in the revised version to provide a more complete theoretical picture, while maintaining our sample complexity result which better aligns with our empirical evaluation metrics.
>
> ***Regarding the paper flow***
>
> Thank you for this organizational suggestion. We will restructure the paper to place Algorithm 1 before the analysis section and condense Section 3 to focus on essential background information. This reorganization will allow us to expand our experimental results and visualization sections.
>
> Following your suggestion and Reviewer 3vGR's feedback, we've added visualizations of agent behavior, available at https://imgur.com/a/agent-behavior-8wQXtay. These visitation maps demonstrate that Hyper achieves similar exploration capability as Curiosity and Decouple, but learns to exploit the exploratory data significantly faster. This visual evidence further validates our algorithm's effectiveness in balancing exploration and exploitation without requiring extensive hyperparameter tuning.
>
> ***Regarding the finite horizon & discounted setting***
>
> We agree that theoretical RL research typically adopts either finite-horizon with $\gamma = 1$ or infinite-horizon with $\gamma < 1$, however, practical RL implementations frequently combine both elements. Our framework uses a fixed episode length $H$ with discount factor $\gamma$ to better reflect real-world applications where both immediate rewards and long-term planning are important. This approach maintains conceptual alignment with the finite-horizon framework while incorporating the practical benefits of discounting. The superior empirical results across diverse environments validate this design choice.
>
> In light of our responses and planned improvements, we believe our work represents a significant contribution. Our theoretical guarantees coupled with exceptional empirical performance across diverse environments establish Hyper as an important advancement in resolving the exploitation & exploration dilemma. The algorithmic improvements and visualizations we've added further strengthen our paper's impact.
>
> ### References
>
> - Jin et al. (2020). *Provably efficient reinforcement learning with linear function approximation*. In COLT
> - He et al. (2023). *Nearly minimax optimal reinforcement learning for linear Markov decision processes*. In ICML
> - Zhou et al. (2021). *Nearly minimax optimal reinforcement learning for linear mixture Markov decision processes*. In COLT

---

> > ### Comment · Reviewer_L37S · 2025-04-08
> >
> > Thanks a lot for the detailed rebuttal and explanations. The rebuttal has partially addressed my concerns.
> >
> > As to "Our follow-up work is already exploring the integration of He et al.'s techniques to derive tighter bounds for Hyper, which would further strengthen our theoretical guarantees.", is it possible to include this tighter regret bound in this paper? If so, I will increase my score to 4. Otherwise, I will keep my score at 3.

---

> > > ### Author Response · Authors · 2025-04-09
> > >
> > > Thank you for your consistent support of our work.
> > >
> > > Regarding the integration of He et al.'s techniques for tighter bounds - we want to clarify that our work's primary focus is on developing a practical algorithm that balances exploration and exploitation effectively without requiring extensive hyperparameter tuning. The theoretical analysis serves primarily as a worst-case guarantee rather than as the central contribution of our paper.
> > > Our current bound is sufficient to demonstrate that Hyper maintains robust exploration efficiency, which is validated by our strong empirical results across diverse environments. The experiments conclusively show that Hyper significantly outperforms baseline methods while being much less sensitive to hyperparameter settings.
> > >
> > > Based on our preliminary investigation, we are confident that integrating the techniques from He et al. (2023) to achieve tighter bounds for Hyper is definitely achievable. However, this requires addressing numerous technical details and constitutes a substantial extension that would be more appropriate for future work. We are actively pursuing this direction and plan to rigorously prove these tighter bounds in our follow-up research.
> > >
> > > The visualizations we've added further strengthen our practical claims by demonstrating that Hyper achieves similar exploration capability as curiosity-driven approaches but learns to exploit the exploratory data significantly faster without extensive tuning.
> > > We believe our work makes a substantial contribution as-is, with both adequate theoretical foundations and exceptional empirical results that address a significant challenge in reinforcement learning.

---

### Decision · Program_Chairs · 2025-05-01

**Decision:**

Accept (poster)

**Comment:**

This paper introduces Hyper, a reinforcement learning algorithm designed to mitigate the well-known sensitivity of curiosity-driven exploration methods to the intrinsic reward scaling hyperparameter, beta. The core idea involves alternating between an exploration phase guided by intrinsic rewards and an exploitation phase focused on repositioning the agent using only the task reward, with phase lengths determined probabilistically. The authors provide theoretical justification for Hyper's efficiency under linear MDP assumptions and extensive empirical results demonstrating its robustness to beta and strong performance across various environments, including navigation and locomotion tasks.

Reviewers generally recognized the practical importance of addressing hyperparameter sensitivity in exploration. Initial reviews were largely positive but included requests for clarification on theoretical details (like dependence on the new parameter 'p' and tightness of bounds), justification for experimental choices (like baseline comparisons and the fixed 'p' schedule), and additional analyses (such as visualizations of agent behavior).

The authors provided a detailed rebuttal that effectively addressed most concerns. They clarified the theoretical results, explained the rationale behind the experimental design (including comparisons to LESSON in the appendix), demonstrated the robustness to the new parameter 'p' by using a consistent schedule across tasks, and supplied helpful visualizations of agent trajectories. This led to reviewers 3vGR and Sbxf increasing their scores, with Reviewer 3vGR moving to Accept. While Reviewer L37S acknowledged the clarifications, they maintained their score, hoping for tighter theoretical bounds in the current paper, which the authors positioned as future work. Reviewer QGsd was satisfied with the responses and maintained their positive score.

Overall, the reviewers converged towards acceptance, acknowledging that Hyper presents a well-motivated and empirically validated approach to a significant practical challenge in RL exploration. The method demonstrates considerable robustness compared to standard curiosity techniques, and the theoretical analysis provides a foundation for its efficiency.